# RETROFORMER: RETROSPECTIVE LARGE LANGUAGE AGENTS WITH POLICY GRADIENT OPTIMIZATION

**Weiran Yao[†], Shelby Heinecke[†], Juan Carlos Niebles[†], Zhiwei Liu[†], Yihao Feng[†], Le Xue[†], Rithesh Murthy[†], Zeyuan Chen[†], Jianguo Zhang[†], Devansh Arpit[†], Ran Xu[†], Phil Mui[†], Huan Wang[†, *], Caiming Xiong[†, *], Silvio Savarese[†, *]**

[†]Salesforce AI Research

## ABSTRACT

Recent months have seen the emergence of a powerful new trend in which large language models (LLMs) are augmented to become autonomous language agents capable of performing objective oriented multi-step tasks on their own, rather than merely responding to queries from human users. Most existing language agents, however, are not optimized using environment-specific rewards. Although some agents enable iterative refinement through verbal feedback, they do not reason and plan in ways that are compatible with gradient-based learning from rewards. This paper introduces a principled framework for reinforcing large language agents by learning a retrospective model, which automatically tunes the language agent prompts from environment feedback through policy gradient. Specifically, our proposed agent architecture learns from rewards across multiple environments and tasks, for fine-tuning a pre-trained language model which refines the language agent prompt by summarizing the root cause of prior failed attempts and proposing action plans. Experimental results on various tasks demonstrate that the language agents improve over time and that our approach considerably outperforms baselines that do not properly leverage gradients from the environment.

## 1 INTRODUCTION

Recently, we have seen the emergence of a powerful new trend in which large language models (LLMs) are augmented to become autonomous language action agents capable of performing tasks on their own, ultimately in the service of a goal, rather than responding to queries from human users. Prominent studies, including ReAct (Yao et al., 2023), Toolformer (Schick et al., 2023), Hugging-GPT (Shen et al., 2023), Generative Agents (Park et al., 2023), WebGPT (Nakano et al., 2021), AutoGPT (Gravitas, 2023), BabyAGI (Nakajima, 2023), and Langchain (Chase, 2023), have successfully showcased the viability of creating autonomous decision-making agents by leveraging the capabilities of LLMs. These approaches use LLMs to generate text-based outputs and actions that can be further employed for making API calls and executing operations within a given environment.

Given the immense scale of LLMs with an extensive parameter count, the behaviors of most existing language agents, however, are not optimized or aligned with environment reward functions. An exception is a very recent language agent architecture, namely Reflexion (Shinn et al., 2023), and several other related work, e.g., Self-Refine (Madaan et al., 2023b) and Generative Agents (Park et al., 2023), which use verbal feedback, namely self-reflection, to help agents learn from prior failure. These reflective agents convert binary or scalar reward from the environment into verbal feedback in the form of a textual summary, which is then added as additional context to the prompt for the language agent. The self-reflection feedback acts as a semantic signal by providing the agent with a concrete direction to improve upon, helping it learn from prior mistakes and prevent repetitive errors to perform better in the next attempt.

Although the self-reflection operation enables iterative refinement, generating useful reflective feedback from a pre-trained, frozen LLM is challenging, as showcased in Fig. 1, since it requires the

---

[*]Corresponding Authors
[†]Website for **Retroformer** & demos: https://Retroformer.github.io/
[‡]Code: https://github.com/SalesforceAIResearch/Retroformer

LLM to have a good understanding of where the agent made mistakes in a specific environment, i.e., the credit assignment problem (Sutton & Barto, 2018), as well as the ability to generate a summary containing actionable insights for improvement. The verbal reinforcement cannot be optimal, if the frozen language model has not been properly fine-tuned to specialize in credit assignment problems for the tasks in given environments. Furthermore, the existing language agents do not reason and plan in ways that are compatible with differentiable, gradient-based learning from rewards by exploiting the existing abundant reinforcement learning techniques. To address these limitations, this paper introduces **Retroformer**, a principled framework for reinforcing language agents by learning a plug-in retrospective model, which automatically refines the language agent prompts from environment feedback through policy optimization. Specifically, our proposed agent architecture can learn from arbitrary reward information across multiple environments and tasks, for iteratively fine-tuning a pre-trained language model, which refines the language agent prompts by reflecting on failed attempts and assigning credits of actions taken by the agent on future rewards.

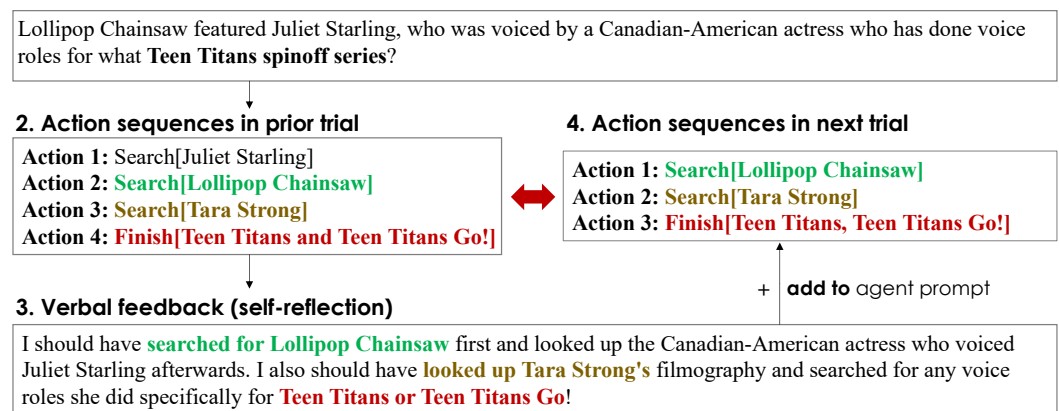

Figure 1: An example of uninformative self-reflections from a frozen LLM. The root cause of failure in prior trial is that the agent should have only submitted the spinoff series "Teen Titans Go" and not "Teen Titans" in the answer. The agent forgot its goal during a chain of lengthy interactions. The verbal feedback from a frozen LLM, however, only rephrases the prior failed actions sequences as the proposed plan, resulting repetitive, incorrect actions in the next trial.

We conduct experiments on a number of real-world tasks including HotPotQA (Yang et al., 2018), which involves search-based question answering tasks, AlfWorld (Shridhar et al., 2021), in which the agent solves embodied robotics tasks through low-level text actions, and WebShop (Yao et al., 2022), a browser environment for web shopping. We observe **Retroformer** agents are faster learners compared with Reflexion, which does not use gradient for reasoning and planning, and are better decision-makers and reasoners. More concretely, **Retroformer** agents improve the success rate in HotPotQA by 18% with 4 retries, 36% in AlfWorld with 3 retries and 4% in WebShop, which demonstrate the effectiveness of gradient-based learning for LLM action agents.

To summarize, our contributions are the following:

- The paper introduces **Retroformer**, which iteratively refines the prompts given to large language agents based on environmental feedback to improve learning speed and task completion. We take a policy gradient approach with the Actor LLM being part of the environment, allowing learning from a wide range of reward signals for diverse tasks.

- The proposed method focuses on fine-tuning the retrospective model in the language agent system architecture, without accessing the Actor LLM parameters or needing to propagate gradients through it. The agnostic nature of **Retroformer** makes it a flexible plug-in module for various types of cloud-based LLMs, such as OpenAI GPT or Google Bard.

## 2 RELATED WORK

**Autonomous Language Agents**   We summarize in Table 1 the recent language agent literature related to our work from five perspectives and differentiate our method from them. The completion of a complex task typically involves numerous stages. An AI agent must possess knowledge of these stages and plan accordingly. Chain-of-Thoughts or CoT (Wei et al., 2022) is the pioneering work that prompts the agent to decompose challenging reasoning tasks into smaller, more manageable steps. ReAct (Yao et al., 2023), on the other hand, proposes the exploitation of this reasoning and acting proficiency within LLM to encourage interaction with the environment (e.g. using the Wikipedia search API) by mapping observations to the generation of reasoning and action traces or API calls in natural language. This agent architecture has spawned various applications, such as HuggingGPT (Shen et al., 2023), Generative Agents (Park et al., 2023), WebGPT (Nakano et al., 2021), AutoGPT (Gravitas, 2023), and BabyAGI (Nakajima, 2023).

Table 1: Related work on large language agents.

| Approach | Gradient learning | Arbitrary reward | Iterative refinement | Hidden constraints | Decision making | Memory |
|---|---|---|---|---|---|---|
| CoT (Wei et al., 2022) | ✗ | ✗ | ✗ | ✗ | ✗ | ✗ |
| ReAct (Yao et al., 2023) | ✗ | ✗ | ✗ | ✓ | ✓ | ✓ |
| Self-refine (Madaan et al., 2023b) | ✗ | ✗ | ✓ | ✗ | ✗ | ✗ |
| RAP (Hao et al., 2023) | ✗ | ✗ | ✓ | ✓ | ✓ | ✓ |
| Reflexion (Shinn et al., 2023) | ✗ | ✗ | ✓ | ✓ | ✓ | ✓ |
| **Retroformer** (our method) | ✓ | ✓ | ✓ | ✓ | ✓ | ✓ |

However, these approaches fail to learn from valuable feedback, such as environment rewards, to enhance the agent's behaviors, resulting in performances that are solely dependent on the quality of the pre-trained LLM. Self-refine (Madaan et al., 2023a) addresses this limitation by employing a single LLM as a generator, refiner, and provider of feedback, allowing for iterative refinement of outputs. However, it is not specifically tailored for real-world task-based interaction with the environment. On the other hand, RAP (Hao et al., 2023) repurposes the LLM to function as both a world model and a reasoning agent. It incorporates Monte Carlo Tree Search for strategic exploration within the extensive realm of reasoning with environment rewards. This approach enables effective navigation and decision-making in complex domains. Recently, Shinn et al. (2023) presents Reflexion, a framework that equips agents with dynamic memory and self-reflection capabilities, enhancing their reasoning skills. Self-reflection plays a pivotal role, allowing autonomous agents to iteratively refine past actions, make improvements, and prevent repetitive errors.

**Transformer Reinforcement Learning**   Reinforcement learning with a provided reward function or a reward-labeled dataset, commonly referred to as RLHF, has become a standard practice within the LLM fine-tuning pipeline. These endeavors have convincingly demonstrated the efficacy of RL as a means to guide language models towards desired behaviors that align with predefined reward functions encompassing various domains, including machine translation, summarization, and generating favorable reviews. Among the prevalent transformer RL methods are online RL algorithms such as Proximal Policy Optimization or PPO (Schulman et al., 2017), and offline RL techniques such as Implicit Language Q-Learning or ILQL (Snell et al., 2022) and Direct Preference Optimization or DPO (Rafailov et al., 2023). These methods have been implemented in TRL/TRLX (von Werra et al., 2020; Max et al., 2023) distributed training framework.

## 3 NOTATION AND FORMULATION

In this work, we denote a large language model (LLM) based action agent as a function $\mathcal{M}_{\xi_l} : \mathcal{X} \to \mathcal{A}$, where $\mathcal{X}$ is the space of prompts, which may include the actual prompts $x^u$ provided by the users, as well as some contextual information $c \in \mathcal{C}$. Here $\mathcal{C}$ is the space of context as a representation of the current state $\mathcal{S}$ returned by the environment $\Omega$. $\mathcal{A}$ is the space of actions. Note the actions taken by most language model based agents are sampled auto-repressively, so $\mathcal{M}$ is a random function. The subscript $\xi_l$ denotes the re-parameterized random variables involved in the sampling process. Another note is, the LLM-based agent itself is stateless. All the states and possible memorization are characterized as text in the agent prompt $x$.

The environment is defined as a tuple $(\mathcal{T}_{\xi_o}, \mathcal{R})$. $\mathcal{T}_{\xi_o} : \mathcal{S} \times \mathcal{A} \to \mathcal{S}$ is the state transition function, where $\mathcal{S}$ is the space of states and $\mathcal{A}$ is the action space. Here we assume the states and actions are represented using text. Again we used $\xi_o$ to represent the randomness involved in the state transition. For each state $s \in \mathcal{S}$, a reward function is defined as $\mathcal{R} : \mathcal{S} \to \mathbb{R}$. At each step of the play, the state $s$ is described using natural language, and integrated into the context $c$. In the context, previous states may also be described and embedded to help LLMs making a good guess on the next action to take. As in all the reinforcement learning setting, the final goal is to maximize the cumulative rewards, or episode returns $G_{cum} = \sum_{t=0}^{T} R(s_t)$. In many situations, the rewards are sparse, i.e., $R(s_t)$ are mostly zero except very few states, such as in the terminal state for indicating task success or failure.

The retrospective model takes the all the previous states $s_{1,\cdots,t}$, actions $a_{1,\cdots,t}$, rewards $r_{1,\cdots,t}$, and the user prompt $x^u$ as input, and massage them into a new prompt $x$ to be consumed by the LLM:

$$\Gamma_{\xi_r,\Theta} : [\mathcal{S}_i, \mathcal{A}_i, \mathcal{R}_i, \mathcal{X}_i^u]_{i=1}^t \to \mathcal{X}, \tag{1}$$

where $\xi_r$ stands for the randomness involved in the retrospective model, and $\Theta$ is the set of learnable parameters in the retrospective model. The goal of the RL optimization is

$$\arg\max_{\Theta} \quad \mathbb{E}_{\xi_l,\xi_o,\xi_r}\left[\sum_{t=1}^{T} R(s_t)\right] \quad s.t.$$

$$s_{t+1} = \mathcal{T}_{\xi_o}\left(s_t, \mathcal{L}_{\xi_l} \circ \Gamma_{\xi_r,\Theta}\left([s_i, a_i, r_i, x_i^u]_{i=1}^t\right)\right), \quad \forall t \in \{1, \cdots, T-1\} \tag{2}$$

Note that the only learnable parameters are in the retrospective model $M_r$. Since LLM action agent is frozen, it can be considered as part of the environment. Specifically, if we construct another environment with the transition function $\mathcal{T}' = \mathcal{T}(\mathcal{S}, \bullet) \circ \mathcal{L} : \mathcal{S} \times \mathcal{X} \to \mathcal{S}$, and the same reward function $\mathcal{R}$, then Eq. (2) is just a regular RL optimization so all the popular RL algorithms apply.

## 4 OUR APPROACH: REINFORCING RETROSPECTIVE LANGUAGE AGENT

As illustrated in Fig. 2, our proposed framework **Retroformer** is comprised of two language model components: an **actor** LLM, denoted as $M_a$, which generates reasoning thoughts and actions, and a **retrospective** LLM, denoted as $M_r$, which generates verbal reinforcement cues to assist the actor in self-improvement by refining the actor prompt with reflection responses.

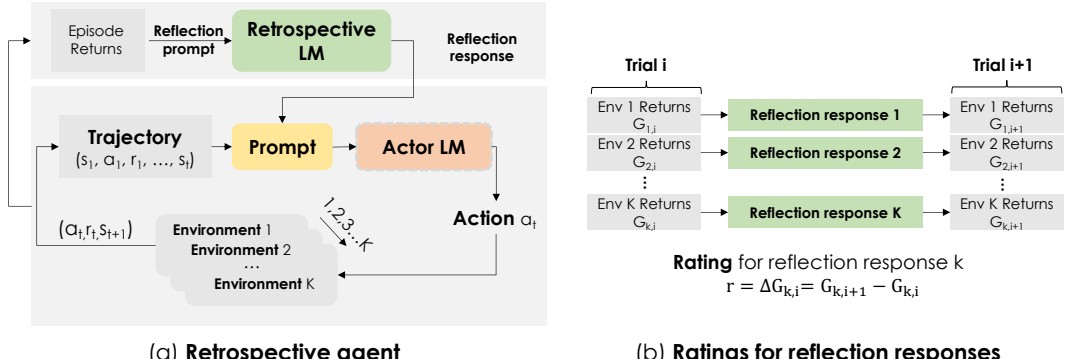

(a) **Retrospective agent**          (b) **Ratings for reflection responses**

Figure 2: Framework overview. (a) The retrospective agent system (Sec. 4.1) contains two LLMs communicating to refine agent prompts with environment feedback. (b) The retrospective LM is fine-tuned with response ratings using proximal policy optimization (Sec. 4.2).

We assume in this paper that the actor model is a frozen LLM whose model parameters are inaccessible (e.g., OpenAI GPT) and the retrospective model is a smaller, local language model that can be fine-tuned under low-resource settings (e.g., Llama-7b). In addition, **Retroformer** has an iterative policy gradient optimization step which is specifically designed to reinforce the retrospective model with gradient-based approach. We provide in this section a detailed description

of each of these modules and subsequently elucidate their collaborative functioning within the **Retroformer** framework. The implementation details are presented in Appendix C.

## 4.1 RETROSPECTIVE AGENT ARCHITECTURE

As illustrated in Fig. 2(a), for the actor and retrospective models, we apply a standard communication protocol modified from the Relexion agent architecture (Shinn et al., 2023), in which the retrospective model refines the actor prompt by appending verbal feedback to the prompt.

**Actor Model** The actor model is a LLM hosted in the cloud, whose model parameters are hidden and frozen all the time. The actor LM is instructed to generate actions with required textual content, taking into account the observed states. Similar to reinforcement learning, we select an action or generation, denoted as $a_t$, from the current policy $\pi_\theta$ at time step $t$ and receive an observation, represented by $s_t$, from the environment. We use ReAct (Yao et al., 2023) as our actor prompt.

$$a_{k,i,t} = M_a \left( [s_{k,i,\tau}, a_{k,i,\tau}, r_{k,i,\tau}]_{\tau=1}^{t-1}, s_{k,i,t} \right). \tag{3}$$

**Retrospective Model** The retrospective model $M_r$ is instantiated as a local LM. Its primary function is to produce self-reflections, offering valuable feedback for diagnosing a possible reason for prior failure and devising a new, concise, high-level plan that aims to mitigate same failure. Operating under a sparse reward signal, such as binary success status (success/failure), the model detects the root cause of failure by considering the current trajectory alongside its persistent memory.

$$y_{k,i} = M_r(\underbrace{[s_{k,i,\tau}, a_{k,i,\tau}, r_{k,i,\tau}]_{\tau=1}^T, G_{k,i}}_{\text{Reflection prompt } x_{k,i}}). \tag{4}$$

This self-reflection feedback $y_{k,i}$ is appended to the actor prompt to prevent repetitive errors in a specific environment in future attempts. Consider a multi-step task, wherein the agent failed in the prior trial. In such a scenario, the retrospective model can detect that a particular action, denoted as $a_t$, led to subsequent erroneous actions and final failure. In future trials, the actor LM can use these self-reflections, which are appended to the prompt, to adapt its reasoning and action steps at time $t$, opting for the alternative action $a_t'$. This iterative process empowers the agent to exploit past experiences within a specific environment and task, thereby avoiding repetitive errors.

**Memory Module** The actor model generates thoughts and actions, by conditioning on its recent interactions (short-term memory) and reflection responses (long-term memory) in the text prompt.

- *Short-term memory*. The trajectory history $\tau_i$ of the current episode $i$ serves as the short-term memory for decision making and reasoning.
- *Long-term memory*. The self-reflection responses that summarize prior failed attempts are appended to the actor prompt as the long-term memory.

To facilitate policy optimization in Section 4.2, we store the instructions and responses of the retrospective model of each trial, together with the episode returns in a local dataset, which we call *replay buffer*. We sample from the replay buffer to fine-tune the retrospective model. The long and short-term memory components provide context that is specific to a given task over several failed trials and the replay buffer provides demonstrations of good and bad reflections across the tasks and environments, so that our **Retroformer** agent not only exploits lessons learned over failed trials in the current task, but also explores by learning from success in other related tasks.

- *Replay buffer*. The memory $D_{\text{RL}}$ which stores the triplets $(x_{k,i}, y_{k,i}, G_{k,i})$ of the reflection instruction prompt $x_{k,i}$, reflection response $y_{k,i}$ and episode return $G_{k,i}$ of trial $i$ and task $k$.

**Reward Shaping** Instead of exactly matching the ground truth to produce a binary reward, we use soft matching (e.g., f1 score) whenever possible to evaluate the alignment of the generated output with the expected answer or product as the reward function. The details are in Appendix C.3.

## 4.2 POLICY GRADIENT OPTIMIZATION

The actor model $M_a$ is regarded as an frozen LLM, such as GPT, with inaccessible model parameters. In this scenario, the most direct approach to enhancing actor performance in a given environment is by refining the actor LM's prompt. Consequently, the retrospective model $M_r$, a smaller local language model, paraphrases the actor's prompt by incorporating a concise summary of errors and valuable insights from failed attempts. We therefore aim to optimize the $M_r$ model using environment reward. The desired behavior of $M_r$ is to improve the actor model $M_a$ in next attempt. Hence, the difference in episode returns between two consecutive trials naturally serves as a reward signal for fine-tuning the retrospective model $M_r$ with reinforcement learning.

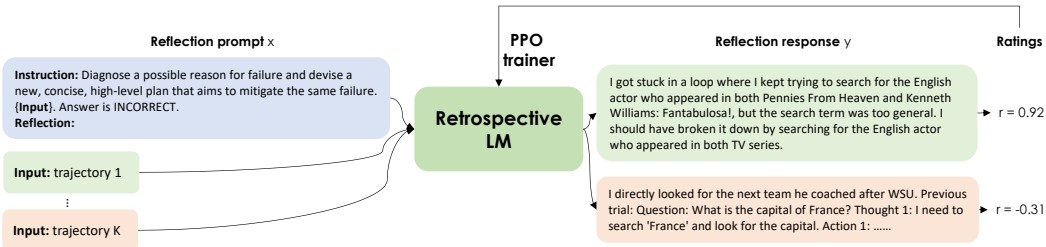

Figure 3: Policy gradient optimization of retrospective LM using RLHF training pipeline.

**Instruction and Response Generation**  The retrospective model generates a pair of instruction and response at the end of each episode $i$ in the environment $k$. In the episode $i$, the actor produces a trajectory $\tau_i$ by interacting with the environment. The reward function then produces a score $r_i$. At the end of the episode, to produce verbal feedback for refining the actor prompt, $M_r$ takes the set of $\{\tau_i, r_i\}$ as the instruction $x_{k,i}$ and is prompted to produce a reflection response $y_{k,i}$. All these instruction-response pairs $(x_{k,i}, y_{k,i})$ across tasks and trials are stored to a local dataset $D_{\text{RL}}$, which we call "replay buffer", for fine-tuning the $M_r$.

**Response Rating**  As illustrated in Fig. 2(b), let us assume a reflection prompt $x_{k,i}$ and the corresponding episode return $G_{k,i}$, and the retrospective model $M_r$ generates the response $y_{k,i}$ that summarizes the mistakes in $i$, which results in the return $G_{k,i+1}$ in the next attempt $i+1$. Because the actor is a frozen LM and the temperature is low as default (Yao et al., 2023), the injected randomness that leads to differences in returns $\Delta G_{k,i} = G_{k,i+1} - G_{k,i}$ are mostly from the reflection responses $y_{k,i}$, in which positive $\Delta G_{k,i}$ indicates better responses that help the actor learn from prior errors, and hence should be rated with higher scores; negative or zero $\Delta G_{k,i}$ indicates worse responses that needs to be avoided and hence should be rated with lower scores. Therefore, we approximate the rating score of a reflection instruction-response pair $(x_{k,i}, y_{k,i})$ as:

$$r(x_{k,i}, y_{k,i}) \triangleq G_{k,i+1} - G_{k,i}. \tag{5}$$

**Proximal Policy Optimization**  The optimization step of **Retroformer** is visualized in Fig. 3. We use the differences of episode returns as the ratings of the generated reflection responses. The retrospective language model is fine-tuned with the response ratings following the RLHF training procedures (although we do not have human in the loop) with proximal policy optimization (PPO):

$$\mathcal{L}_{\text{PPO}} = \mathbb{E}_{x \sim D_{\text{RL}}} \mathbb{E}_{y \sim \text{LLM}_\phi^{\text{RL}}(x)} \left[ r_\theta(x, y) - \beta \log \frac{\text{LLM}_\phi^{\text{RL}}(y|x)}{\text{LLM}^{\text{Ref}}(y|x)} \right], \tag{6}$$

where $(x, y)$ are sampled from the replay buffer (note there is only 1 step in the Retrospective model's trajectory), $r_\theta(x, y)$ is the defined reward model, and the second term in this objective is the KL divergence to make sure that the fine-tuned model $\text{LLM}^{\text{RL}}$ does not stray too far from the frozen reference model $\text{LLM}^{\text{Ref}}$.

For offline training, we collected the dataset $D_{\text{RL}}$ by rolling out a base policy, i.e., the frozen actor LM and the initialized retrospective LM, in the tasks in the training sets for $N$ trials and compute

the ratings. We apply the standard RLHF pipeline to fine-tune the retrospective model offline before evaluating the agent in the validation tasks. In online execution, we use best-of-$n$ sampler, with the scores evaluated by the learned reward model from RLHF pipeline (Ouyang et al., 2022), for generating better retrospective responses in each trial.

## 5 EXPERIMENTS

Extensive experiments are conducted to evaluate our method, including comparisons with ReAct and Reflexion performances, and visualization and discussion of agent's generated text and actions.

### 5.1 EXPERIMENT SETUP

#### 5.1.1 ENVIRONMENT

We use open-source environments: HotPotQA (Yang et al., 2018), WebShop (Yao et al., 2022) and AlfWorld (Shridhar et al., 2021) , which evaluates the agent's reasoning and tool usage abilities for question answering reasoning, multi-step decision making, and web browsing.

**HotPotQA**   The agent is asked to solve a question answering task by searching in Wikipedia pages. At each time step, the agent is asked to choose from three action types or API calls:

1. SEARCH[ENTITY], which searches the exact entity on Wikipedia and returns the first paragraph if it exists. If not, it will return some similar entities to search.
2. LOOKUP[KEYWORD], which returns the next sentence containing keyword in the last passage successfully found by Search.
3. FINISH[ANSWER], which returns the answer and finishes the task.

**AlfWorld**   The agent is asked to perform six different tasks, including finding hidden objects (e.g., finding a spatula in a drawer), moving objects (e.g., moving a knife to the cutting board), and manipulating objects with other objects (e.g., chilling a tomato in the fridge) by planning with the following action APIs, including GOTO[LOCATION], TAKE[OBJ], OPEN[OBJ], CLOSE[OBJ] , TOGGLE[OBJ], CLEAN[OBJ], HEAT[OBJ], and COOL[OBJ], etc.

**WebShop**   The agent is asked to solve a shopping task by browsing websites with detailed product descriptions and specifications. The action APIs include searching in the search bar, i.e., SEARCH[QUERY] and clicking buttons in the web pages, i.e., CHOOSE[BUTTON]. The clickable buttons include, product titles, options, buy, back to search, prev/next page, etc.

### 5.2 EXPERIMENT SETTINGS

We use GPT-3 (model: text-davinci-003) and GPT-4 as the frozen actor model. For the retrospective model, we fine-tune it from LongChat (model: longchat-7b-16k). The implementation details, which include data collection and model training are in Appendix C.

**Evaluation Metrics**   We report the success rate over validation tasks in an environment. The agent is evaluated on 100 validation tasks from the distractor dev split of open-source HotPotQA dataset, 134 tasks in AlfWorld and 100 tasks in WebShop, as in (Shinn et al., 2023).

**Baselines**   We experiment with two language agent baselines: **1) ReAct (Yao et al., 2023)**. This is the state-of-the-art frozen language agent architecture, which does not learn from the environment rewards at all, thus serving as a baseline for showing how the agent performs without using environment feedback. **2) Reflexion (Shinn et al., 2023)**. This is the state-of-the-art language agent architecture that the authors identify from literature so far. This agent enhances from verbal feedback of the environment, but does not use gradient signals explicitly. It can serve as a baseline for showing the effectiveness of gradient-based learning. **3) SAC.** Furthermore, we include one online RL algorithm, i.e., Soft Actor-Critic (Haarnoja et al., 2018), or SAC as baseline model for comparison.

5.3 RESULTS

We present the experiment results in Table 2 and discuss the details below.

Table 2: Results with **Retroformer** in the HotPotQA, AlfWorld and Webshop environments. We report the average success rate for the language agents over tasks in the environment. "#Params" denotes the learnable parameters of each approach. "#Retries" denotes the number of retry attempts. "LoRA $r$" denotes the rank of low-rank adaptation matrices for fine-tuning.

| Method | #Params | #Retries | HotPotQA | | AlfWorld | | WebShop | |
|---|---|---|---|---|---|---|---|---|
| SAC | 2.25M | N=1 | 27% | | 58.95% | | 30% | |
| | | N=4 | 27% | | 59.7% | | 30% | |
| **Actor LLM** | | | | | | | | |
| | | | GPT-3 | GPT-4 | GPT-3 | GPT-4 | GPT-3 | GPT-4 |
| ReAct | 0 | | 34% | 40% | 62.69% | 77.61% | 33% | 42% |
| Reflexion | 0 | N=1 | 42% | 46% | 76.87% | 81.34% | 35% | 42% |
| | | N=4 | 50% | 52% | 84.33% | 85.07% | 35% | 44% |
| **Retroformer** (w/ LoRA r=1) | 0.53M | N=1 | 45% | 48% | 93.28% | 95.62% | 36% | 43% |
| | | N=4 | 53% | 53% | 100% | 100% | 36% | 45% |
| **Retroformer** (w/ LoRA r=4) | 2.25M | N=1 | 48% | 51% | 97.76% | 97.76% | 34% | 43% |
| | | N=4 | 54% | 54% | 100% | 100% | 36% | 46% |

**Question Answering – HotPotQA** We visualize the performances of **Retroformer** against the baselines in Fig. 4. As shown in Table 2, we observe that our method consistently improve the agent performances over trials and the effects of fine-tuned retrospective model (**Retroformer**) are mostly significant in the first few trials.

Furthermore, as shown in Fig. 4, our agent outperforms the two strong baselines. Specifically, the results indicate that our reinforced model provides the language agents with better reflection responses in early trials, which enables the agents to learn faster, while also achieving better performances in the end. Our **Retroformer** agent achieves 54% success rate in 4 trials, which is better than the state-of-the-art 50% success rate reported in (Jang, 2023) that uses a much larger frozen language model, i.e., GPT-3 (model: text-davinci-003) as the reflection component. The results show the effectiveness of our policy gradient approach for fine-tuning the agent with offline samples.

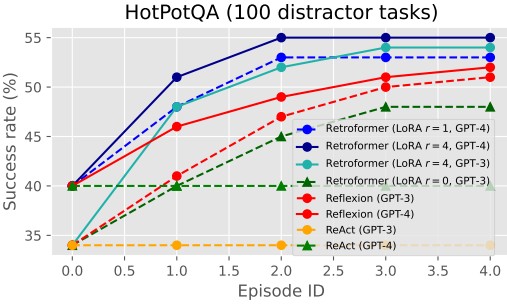

Figure 4: **Retroformer** shows faster and consistent performance improvement of success rate.

We then examine how the retrospective model is improved with policy optimization by comparing the generated responses from the frozen LM and the ones from the fine-tuned, reinforced LM. As an example, Fig. 5 illustrates how the uninformative self-reflections from a frozen LLM, which we propose in Fig. 1, are tackled by RL. The agent failed in the last attempt because it submitted "Teen Tians" and "Teen Titans Go" as the answer to the Teen Titans spin-off series, while the correct answer includes only "Teen Titans Go"; The agent forgot its original goal during a chain of lengthy interactions. The self-reflection from the frozen model reiterated the prior action sequences that led to failure as the steps that the agent should have done, which prompts the agent to repeat these steps in the next attempt, resulting in an infinite loop. On the contrary, our reinforced response prompts the agent to focus on **spinoff series** and asks the agent to find the answer in the previous actions and observations with the search results of "Tara Strong". **Retroformer** presents better credit assignment and root cause analysis abilities, and has the power to generate actionable insights.

**Decision Making – AlfWorld** As showcased in Fig. 6(a), the performance improvement by **Retroformer** over the frozen baselines is significant and our method solves the environment

| Question | Prior failed trial | Reflection (Green: Retroformer; Brown: Frozen LM) |
|---|---|---|
| Lollipop Chainsaw featured Juliet Starling, who was voiced by a Canadian-American actress who has done voice roles for what Teen Titans **spinoff series**? | **Thought 1:** I need to search Juliet Starling and find the Canadian-American actress who voiced her, then find the Teen Titans spinoff series she has done voice roles for. **Action 1:** Search[Juliet Starling] **Observation 1:** Could not find [Juliet Starling]. … 

 **Thought 2:** Juliet Starling is a character from Lollipop Chainsaw, so I can search Lollipop Chainsaw. **Action 2:** Search[Lollipop Chainsaw] **Observation 2:** Lollipop Chainsaw is a … 

 **Thought 3:** Juliet Starling is voiced by Tara Strong, so I need to search Tara Strong and find the Teen Titans spinoff series she has done voice roles for. **Action 3:** Search[Tara Strong] **Observation 3:** Tara Lyn Strong… 

 **Thought 4:** Tara Strong has voiced characters in Teen Titans and Teen Titans Go!, so the answer is Teen Titans and Teen Titans Go!. **Action 4:** Finish[Teen Titans and Teen Titans Go!] | I failed to find the Teen Titans spinoff series that Tara Strong has done voice roles for. Instead, I found Lollipop Chainsaw, a video game that Juliet Starling appeared in. I should have searched Tara Strong and Teen Titans or Teen Titans Go! specifically in my previous action to find the correct answer. 

 My thought process was strong but my execution was flawed. I should have searched for Lollipop Chainsaw first and looked up the Canadian-American actress who voiced Juliet Starling afterwards. I also should have looked up Tara Strong's filmography and searched for any voice roles she did specifically for Teen Titans or Teen Titans Go!.Next trial:Question: The novel was adapted into a film by a director known for his work on another iconic 1980s franchise. What novel is this film based on?Thought 1: |

Figure 5: Response refinement from the reinforced retrospective model. Note that the lengthy observation step in the prior failed trial column is abbreviated for better presentation purposes.

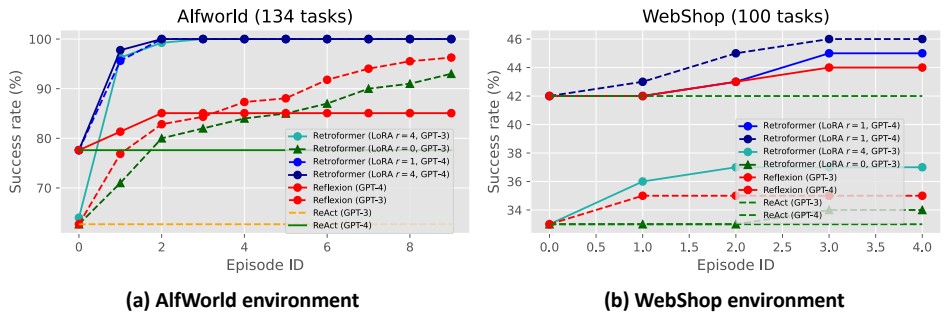

**(a) AlfWorld environment**    **(b) WebShop environment**

Figure 6: Comparisons of **Retroformer** against baselines in (a) AlfWorld and (b) WebShop environments under different base Actor LLM and LoRA rank $r = 1, 4$.

within 3 retries. Similar patterns are observed that the agent performs slightly better with more learnable parameters ($r = 4$) and that the improvements are mostly from early retries. We find that the reinforced retrospective model behaves like a summarization model of the prior failed plans and finds the differences of the prior plan with the task descriptions. With the permissible actions seen in the task instructions, this behavior effectively prevents repetitive failures and reduces search spaces.

**Web Browsing – WebShop** As in Fig. 6(b), the performance improvement by **Retroformer** over the frozen baselines is observed but the improvements may be limited, when compared with HotPotQA and AlfWorld, with 4% improvement in success rate with 4 retries. This limitation was also observed in (Shinn et al., 2023) as web browsing requires a significant amount of exploration with more precise search queries, if compared with HotPotQA. The results probably indicate that the verbal feedback approach (Reflexion, Retroformer) is not an optimal method for this environment, but our fine-tuning method still proves effective.

## 6 CONCLUSION

In this study, we present **Retroformer**, an elegant framework for iteratively improving large language agents by learning a plug-in retrospective model. This model, through the process of policy optimization, automatically refines the prompts provided to the language agent with environmental feedback. Through extensive evaluations on real-world datasets, the method has been proven to effectively improve the performances of large language agents over time both in terms of learning speed and final task completion.

By considering the LLM action agent as a component of the environment, our policy gradient approach allows learning from arbitrary reward signals from diverse environments and tasks. This facilitates the iterative refinement of a specific component within the language agent architecture – the retrospective model, in our case, while circumventing the need to access the Actor LLM parameters or propagate gradients through it. This agnostic characteristic renders **Retroformer** a concise

and adaptable plug-in module for different types of cloud-hosted LLMs, such as OpenAI GPT and Bard. Furthermore, our approach is not limited to enhancing the retrospective model alone; it can be applied to fine-tune other components within the agent system architecture, such as the memory and summarization module, or the actor prompt. By selectively focusing on the component to be fine-tuned while keeping the remainder fixed, our proposed policy gradient approach allows for iterative improvements of the component with reward signals obtained from the environment.

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

*Appendix for*

## "Retroformer: Retrospective Large Language Agents with Policy Gradient Optimization"

## A CHALLENGES

Although LLMs are not designed to handle tool use or take actions, it has been observed (Gravitas, 2023; Nakajima, 2023; Chase, 2023) that empirically for text-rich environment, especially when the actions and states are accurately described using natural languages, LLMs work surprisingly well. However there are still plenty of challenges applying LLM-based agents. Here we list several below.

**Spurious Actions**   LLMs are not pre-trained or designed with an action-agent application in mind. Even some restrictions are explicitly specified in the prompt, the LLM model may still generate spurious actions that are not in the action space $\mathcal{A}$.

**Limited Prompt Length**   LLM itself is stateless. However, in applications it is preferred to empower agents with states or memories for better performance. It has been observed that LLM based agents are easy to run into infinite loops if the states are not handled nicely. Many LLM agents concatenate all the previous state descriptions and actions into the prompt so that LLM as a way to bestow "state" to the LLM. Inevitably this methodology runs into the prompt length issues. As the trajectory grows longer, the prompt runs out of spaces.

**Heuristic Prompt Engineering**   Even though a lot of paradigms have been proposed to improve LLM agents' performance (Yao et al., 2023; Ahn et al., 2022), there is a lack of systematic methodologies for consistent model refinement. In fact, manual prompt tuning is still widely used in a lot of the application scenarios.

**Prohibitive Training**   Most of the well-performing LLMs are too large to be fit in just one or two GPUs. It is technically challenging to optimize the LLMs directly as is done in the the classical reinforcement learning setting. In particular, OpenAI has not provided any solution for RL based finetuning. Most of the issues are caused by the fact that LLMs are not pre-trained or designed with an action-agent application in mind.

## B INTUITION

Compared to the LLM-based action agents, classical RL agents, though not able to handle text-based environments as nicely in the zero shot setting, are able to keep improving based on the feedback and rewards provided by the environment. Popular RL algorithms include Policy Gradient (Sutton et al., 2000), Proximal Policy Optimization Algorithm (PPO) (Schulman et al., 2017), Trust Region Policy Optimization (TRPO) (Schulman et al., 2015), and Advantage Actor Critic methods (Mnih et al., 2016).

In this draft we are proposing a simple but powerful novel framework to tackle the challenges mentioned above. On one hand, we would like to leverage the classical RL based optimization algorithms such as policy gradient to improve the model performance. On the other hand, our framework avoids finetuning on the LLM directly. The key is, instead of training the LLM directly, we train a retrospective LM. The retrospective LM takes users' prompt, rewards and feedback from the environment as input. Its output will be prompt for the actual LLM to be consumed. RL algorithms are employed to optimize the weights in the retrospective LM model instead of directly on the LLM. In our framework the weights in the actual LLM is assumed to be fixed (untrainable), which aligns well with the application scenario when the LLM is either too large to tune or prohibited from any tuning.

Another perspective viewing our framework is, we train a retrospective LM to apply automatic prompt tuning for the LLM agents. In this case, the RL algorithms such as policy gradients are employed to optimize the prompts. Ideally the retrospective LM can help summarize the past "experience", the users' prompt, the environments' feedback into a condensed text with length limit

so that it is easier for the LLM to digest. To some extent, in our setting the original LLM can be considered as part of the environment since its parameters are all fixed.

## C  IMPLEMENTATION DETAILS

### C.1  RETROFORMER

**Model**  We use GPT-3 (model: text-davinci-003) as the frozen actor model. For the retrospective model, we instantiate it from LongChat (model: longchat-7b-16k), which is a LM with 16k context length by fine-tuning llama-7b on instruction-following samples from ShareGPT. In all experiments, we set the temperature of actor LM as zero, i.e., T=0 and top p =1 to isolate the randomness of LM from the effects of reflections. We acknowledge that setting a higher temperature value can encourage exploration but it can obscure the impact of the proposed approaches, making it difficult to compare against existing baselines with T=0 (Yao et al., 2023; Shinn et al., 2023).

**Setup**  Our proposed learning framework is developed by using multiple open-source tools as follows. We use the OpenAI connectors from *langchain* to build our actor models $M_a$. During inference of the retrospective model, we host an API server using *FastChat* and integrates it with *langchain* agents. The tool can host longchat-7b-16k with concurrent requests to speed up RL policy rollouts. For fine-tuning the retrospective model, we develop our training pipeline with *trl*, which supports transformer reinforcement learning with PPO trainer.

We present the details of the specific prompts we used and the full agent demonstrations and examples for each environment in Appendix E.

**Data Collection**  For HotPotQA environment, We collected 3,383 reflection samples by running the base rollout policy for 3 trials ($N = 3$) for 3,000 tasks in the training set, in which 1,084 instruction-response pairs have positive ratings. For AlfWorld, we collected 523 reflection samples and for WebShop, we collected 267 reflection samples.

**Training**  We fine-tune the retrospective model $M_r$ with 4-bit quantized LoRA adapters (r=1 or r=4) on the offline RL datasets with epochs=4; batch size=8; lr=1.4e-5. The number of trainable parameters is 0.53M (0.015% of llama-7b) or 2.25M. Since longchat-16k is based on Llama, we used the default llama recipes for finetuning. Specifically, we first run supervised fine-tuning trainer on the samples with positive ratings for 2 epochs and then the RLHF pipeline, including reward modeling, and RL fine-tuning with PPO, on the whole offline rating dataset using the default settings for llama-7b model. We list the key hyperparameters here:

- **Supervised Finetuning**: learning rate=1e-5, batch size=32, max steps=5,000
- **Reward Modeling**: learning rate=2.5e-5, batch size=32, max steps=20,000
- **Policy Gradient Finetuning**: learning rate=1.4e-5, max steps=20,000, output max length=128, batch size=64, gradient accumulation steps=8, ppo epochs=4

**Reproducibility**  All experiments are done in Google Cloud Platform (GCP) GKE environment with A100 40GB GPUs. The code can be found in `https://anonymous.4open.science/r/Retroformer-F107`. We plan to open source the code repository after the review period.

**Algorithm**  The offline PPO algorithm we used for finetuning the Retrospective component in this paper is presented below in Algorithm 1. It contains three steps: offline data collection, reward model learning, and policy gradient finetuning. We use the offline ratings data to train a reward model first, and plug in the reward model for PPO finetuning.

---

**Algorithm 1** `Retroformer` with Policy Gradient Optimization

---

1: Initialize TEXT-DAVINCI-003 as the Retrospective model with LONGCHAT-16K. Set the maximum trials for rollouts as $N = 3$. The temperature used for sampling $t_s = 0.9$.
2: **Step 1: Offline Data Collection.** Collect multiple rollouts for each environments $k$ ($k = 1, \cdots, K$) for the tasks in the training sets and save as $D_{\text{RL}}$.
3: **for** episode $t = 1, \ldots,$ N **do**
4:     **for** source domain k = 1, . . . , K **do**
5:         Receive trajectory $[s_{k,i,\tau}, a_{k,i,\tau}, r_{k,i,\tau}]_{\tau=1}^T$ and episodic returns $G_{k,i}$ for task $i$.
6:         **for** unsuccessful tasks $j$ **do**
7:             Randomly sample a pair of reflection responses $(y_{k,j}^{(1)}, y_{k,j}^{(2)})$ with Retrospective LM temperature set to $t_s$, with the same instruction prompt defined in Eq. (4).
8:             Roll out the next episode with $y_{k,j}$, and receive the episodic returns $(G_{k,i+1}^{(1)}, G_{k,i+1}^{(2)})$.
9:             Compute reflection response rating by $r(x_{k,i}, y_{k,i}) \triangleq G_{k,i+1} - G_{k,i}$ in Eq. (5).
10:             Label the response with higher ratings as the accepted response while the lower response is labeled as the rejected response.
11:         **end for**
12:     **end for**
13: **end for**
14: **Step 2. Reward Model Learning.** Use the REWARDTRAINER in TRL to train a model for classifying accepted and rejected responses given instructions.
15: **Step 3: Policy Gradient Finetuning.** Plug-in the trained reward model and use the PPOTRAINER in TRL to finetune the Retrospective model for generating reflection responses with higher ratings.

---

## C.2   BASELINE: SOFT-ACTOR CRITIC AGENT

Traditional reinforcement learning methods have been recognized to perform well within the same framework of interaction-feedback-learning. We include one online RL algorithm, i.e., Soft Actor-Critic (Haarnoja et al., 2018), or SAC as baseline model for comparison. Given that the three environments are text-based games, inspired by (Yuan et al., 2018), we do mean-pooling for the embeddings of the generated text outputs, such as "Search[It Takes a Family]" as the agent actions. Therefore, the action space is continuous and is of 768 dimension. We apply LoRA adapters with $r = 4$ on the agent Action model instantiated from longchat-16k, and use SAC to do the online updates, with discount factor gamma=0.99, interpolation factor polyak=0.995, learning rate=0.01, entropy regularzation alpha=0.2, and batch size=8.

## C.3   REWARD FUNCTION

**HotPotQA**   F1 reward is used in the HotPotQA environment for comparing the matching of a generated answer to a question against the ground truth answer. After removing the stopwords in both answers, we calculate the number of common tokens in two answers. Then Precision is # of common tokens divided by # of generated answer tokens and the Recall is # common tokens divided by # ground truth answer tokens. We can then compute f1 from precision and recall.

**AlfWorld**   The binary success (1) and failure (0) of the tasks at the end of episode is used as the reward.

**WebShop**   In each episode, the agent receives a reward $r = \mathcal{R}(s_T, a)$ in the end at timestep $T$, where $a = \texttt{choose[buy]}$, $y$ is the product chosen by the agent in the final state $s_T$, and $Y_{\text{att}}$ and $Y_{\text{opt}}$ are its corresponding attributes and options. The reward is defined as:

$$r = r_{\text{type}} \cdot \frac{|U_{\text{att}} \cap Y_{\text{att}}| + |U_{\text{opt}} \cap Y_{\text{opt}}| + \mathbf{1}[y_{\text{price}} \leq u_{\text{price}}]}{|U_{\text{att}}| + |U_{\text{opt}}| + 1} \tag{7}$$

where the type reward $r_{\text{type}} = \texttt{TextMatch}(\bar{y}, \bar{y}^*)$ is based on text matching heuristics to assign low reward when $y$ and $y^*$ have similar attributes and options but are obviously different types of products. For example, "butter" and "plant-based meat" differ in types but may both contain attributes "cruelty-free", "non-GMO", and an option "size: pack of 2".

# D    ADDITIONAL EXPERIMENTS

| Frozen model response | Reinforced model response |
| --- | --- |
| I got stuck in a loop where I kept searching for the same person, Henry Miller, instead of Beatrice Sylvas Wickens. It's possible that I may have misheard or misunderstood the question. To avoid this, I should have written down the question and made sure I understood it correctly before beginning the search. Additionally, I should have tried searching for "James Henry Miller's wife" instead of just "James Henry Miller". | I think the issue might have been with the search. I was not able to find the name "Miller" in the paragraph, which is why I was not able to find James Henry Miller's wife. I should have searched for "Miller's wife" instead of "James Henry Miller's wife" to avoid this confusion. |
| | New Plan: To avoid confusion, I will search for "Miller's wife" instead of "James Henry Miller's wife" and directly look for the nationality of the wife. |

Figure 7: Response refinement from the reinforced retrospective model with structured format.

**Structured Reflections and Action Plans.**    We discover one emergent behavior of the reinforced model that it can automatically paraphrase the original responses into two separate structured sections, namely *Reflection* section and *New plan:* section, although not being explicitly trained or prompted for. One such example is shown in Fig. 7. The paraphrased response retrospects in the first paragraph and provides actionable insights next, while the response from the frozen LM interleaved both parts in one paragraph, making it hard to comprehend. We can also observer from Fig. 5 that the reinforced response removes the messy, irrelevant "Next trial:" content in the end for cleaner format, which may very likely result from LLM hallucination.

# E    FULL EXAMPLES

## E.1    ACTOR PROMPT ENGINEERING

An example of the HotPotQA actor language model prompt is shown below.

Solve a question answering task with interleaving Thought, Action, Observation steps. Thought can reason about the current situation, and Action can be three types:

(1) Search[entity], which searches the exact entity on Wikipedia and returns the first paragraph if it exists. If not, it will return some similar entities to search.
(2) Lookup[keyword], which returns the next sentence containing the keyword in the last passage successfully found by Search.
(3) Finish[answer], which returns the answer and finishes the task.
You may take as many steps as necessary.
You have attempted to answer the following question before and failed. The following reflection(s) give a plan to avoid failing to answer the question in the same way you did previously. Use them to improve your strategy of correctly answering the given question.

**Reflections:**
- I should have searched for the title of the book instead of the author. I also didn't check for the proverb which was not the question.
- I should have searched for "It Takes a Family by Rick Santorum" instead of just "It Takes a Family" to get the correct answer.
- My search was not concise and I did not specify what I was looking for. I also did not check similar results. In this case, I should have searched for 'It Takes a Family by Rick Santorum' to find the correct answer.

**Question:** It Takes a Family is a response to this 1996 book that was published by who
**Thought 1:** I need to search for the title of the book, "It Takes a Family", and find who published it.
**Action 1:** Search["It Takes a Family"]
**Observation 1:** Richard John Santorum ( san-TOR-əm; born May 10, 1958) is an American politician, attorney, author, and political commentator who represented Pennsylvania in the United States Senate from 1995 to 2007. He was the Senate's third-ranking Republican during the final six years of his tenure. He also ran unsuccessfully for President of the United States in the 2012 Republican primaries, finishing second to Mitt Romney.Santorum was elected to the United States Senate from Pennsylvania in 1994. He served two terms until losing his 2006 reelection bid. A Roman Catholic, Santorum is a social conservative who opposes abortion and same-sex marriage and embraced a cultural warrior image during his Senate tenure. While serving as a senator, Santorum authored the Santorum Amendment, which would have promoted the teaching of intelligent design. He was a leading sponsor of the 2003 federal law known as the Partial-Birth Abortion Ban Act.In the years following his departure from the Senate, Santorum has worked as a consultant, private practice lawyer, and news contributor. He ran for the Republican nomination in the 2012 U.S. presidential election. Before suspending his campaign on April 10, 2012, Santorum exceeded expectations by winning 11 primaries and caucuses and receiving nearly four million votes, making him the runner-up to eventual nominee Mitt Romney. Santorum ran for president again in 2016, but ended his campaign in February 2016 after a poor showing in the Iowa caucuses. In January 2017, he became a CNN senior political commentator. However, he was terminated from his contract with CNN in May 2021 due to comments he made about Native Americans a few weeks prior which were deemed "dismissive".
**Thought 2:** The paragraph mentions Rick Santorum, so I need to search "It Takes a Family by Rick Santorum".
**Action 2:** Search["It Takes a Family by Rick Santorum"]
**Observation 2:** It Takes a Family is a 2005 book by then Pennsylvania Senator Rick Santorum. The title is a response to the 1996 book It Takes a Village by then-First Lady Hillary Clinton. In the book, Santorum states that the family structure is necessary. He argues that liberal social policies have devastated the family structure and that prevailing socially liberal attitudes have systematically devalued families across the board. He argues that the government should take a proactive role in promoting strong families.
**Thought 3:** It Takes a Family was published by Rick Santorum, so the answer is Rick Santorum.
**Action 3:** Finish[Rick Santorum]
**Observation 3:** Answer is INCORRECT

An example of the AlfWorld actor language model prompt is shown below.

```
You are in the middle of a room. Looking quickly around you, you see a cabinet 10, a cabinet 9, a
cabinet 8, a cabinet 7, a cabinet 6, a cabinet 5, a cabinet 4, a cabinet 3, a cabinet 2, a
cabinet 1, a coffeemachine 1, a countertop 2, a countertop 1, a diningtable 2, a diningtable 1, a
drawer 3, a drawer 2, a drawer 1, a fridge 1, a garbagecan 1, a microwave 1, a sinkbasin 1, a
stoveburner 4, a stoveburner 3, a stoveburner 2, a stoveburner 1, and a toaster 1.
```

**[Few-Shot Demonstrations]**

**Your memory for the task below:**
**Trial 0:** I will try to find a different task to complete. I will look for other objects in the
room and try to complete tasks related to those objects. For example, I could try to find a book
to read or a glass to drink from. I will also try to use the objects I have found in the room in
different ways to see if that helps me complete a task.

**Your task is to:** heat some mug and put it in coffeemachine.
**> think:** To solve the task, I need to find and take a mug, then heat it with stoveburner, then
put it in coffeemachine.
Ok.
**> look**
You are in the middle of a room. Looking quickly around you, you see nothing.
**> think:** First I need to find a mug. A mug is more likely to appear in countertop (1-2),
diningtable (1-2), drawer (1-3), fridge (1), sinkbasin (1), stoveburner (1-4), toaster (1). I can
check one by one, starting with countertop 1.
Ok.
**> look**
You are in the middle of a room. Looking quickly around you, you see nothing.
**> go to countertop 1**
On the countertop 1, you see a butterknife 1, a cup 1, a fork 2, and a mug 1.
**> think: Now I find a mug (1). Next, I need to take it.**
Ok.
**> take mug 1 from countertop 1**
You pick up the mug 1 from the countertop 1.
**> think: Now I take a mug (1). Next, I need to heat it with stoveburner (1-4).**
Ok.
**> go to stoveburner 1**
On the stoveburner 1, you see a pan 2.
**> examine stoveburner 1**
On the stoveburner 1, you see a pan 2.
**> examine stoveburner 1**
On the stoveburner 1, you see a pan 2.
**> examine stoveburner 1**
On the stoveburner 1, you see a pan 2.
**> examine stoveburner 1**
On the stoveburner 1, you see a pan 2.

**STATUS:** FAIL

**New plan:** I was stuck in a loop in which I continually examined stoveburner 1 instead of heating
mug 1 with stoveburner 1. I should have taken mug 1 from countertop 1, then heated it with
stoveburner 1, then put it in coffeemachine 1. It did not help to execute two identical actions
in a row. I will try to execute a different action if I am stuck in a loop again.

An example of the WebShop actor language model prompt is shown below.

```
Webshop
[Few-Shot Demonstrations]

Instruction:
i would like a 3 ounce bottle of bright citrus deodorant for sensitive skin, and price lower than
50.00 dollars
[Search]

Your memory for the task below:
Trial 0: In this attempt, I was unsuccessful. I accidentally bought a product that was more than
my budget of $50. The initial search results were not good.

Action: search[3 ounce bright citrus deodorant sensitive skin]
Observation:
[Back to Search]
Page 1 (Total results: 50)
[Next >]
[B078GWRC1J]
Bright Citrus Deodorant by Earth Mama | Natural and Safe for Sensitive Skin, Pregnancy and
Breastfeeding, Contains Organic Calendula 3-Ounce
$10.99
[B078GTKVXY]
Ginger Fresh Deodorant by Earth Mama | Natural and Safe for Sensitive Skin, Pregnancy and
Breastfeeding, Contains Organic Calendula 3-Ounce
$10.99
[B08KBVJ4XN]
Barrel and Oak - Aluminum-Free Deodorant, Deodorant for Men, Essential Oil-Based Scent, 24-Hour
Odor Protection, Cedar & Patchouli Blend, Gentle on Sensitive Skin (Mountain Sage, 2.7 oz,
2-Pack)
$15.95

Action: think[B078GWRC1J and B078GTKVXY are bright citrus deodorant less then 50 dollars. I can
check B078GWRC1J first.]
Observation: OK.

Action: click[B078GWRC1J]
Observation:
[Back to Search]
[< Prev]
scent [assorted scents][bright citrus][calming lavender][ginger fresh][simply non-scents]
size [travel set (4-pack)][3 ounce (pack of 1)][3-ounce (2-pack)]
Bright Citrus Deodorant by Earth Mama | Natural and Safe for Sensitive Skin, Pregnancy and
Breastfeeding, Contains Organic Calendula 3-Ounce
Price: $10.99
Rating: N.A.
[Description]
[Features]
[Reviews]
[Buy Now]

Action: think[For 3 ounce bottle of bright citrus deodorant for sensitive skin, the item has
options 'bright citrus' and '3 ounce (pack of 1)' and seems good to buy.]
Observation: OK.

Action: click[bright citrus]
Observation: You have clicked bright citrus.

Action: click[3 ounce (pack of 1)]
Observation: You have clicked 3 ounce (pack of 1).

Action: click[Buy Now]
```

## E.2 Retrospective Prompt Engineering

An example of the HotPotQA retrospective instruction prompt is shown below.

You are an advanced reasoning agent that can improve based on self reflection. You will be given a previous reasoning trial in which you were given access to an Docstore API environment and a question to answer. You were unsuccessful in answering the question either because you guessed the wrong answer with Finish[<answer>], or you used up your set number of reasoning steps. In a few sentences, Diagnose a possible reason for failure and devise a new, concise, high level plan that aims to mitigate the same failure. Use complete sentences.

Here are some examples.
**[Few-Shot Demonstrations]**

**Previous trial:**
Question: The 2016 Washington State Cougars were led by the coach who previously helmed which other team?

**Thought:** I need to search Washington State Cougars and find the coach of the 2016 team, then find which other team the coach previously helmed.
**Action:** Search[Washington State Cougars]
**Observation:** The Washington State Cougars (known informally as the Cougs) are the athletic teams that represent Washington State University. Located in Pullman, Washington, WSU is a member of the Pac-12 Conference in NCAA Division I. The athletic program comprises ten women's sports and six men's intercollegiate sports, and also offers various intramural sports.

**Thought:** The paragraph does not mention the coach of the 2016 team. I need to look up "coach" to find the coach of the 2016 team.
**Action:** Lookup[coach]
**Observation:** (Result 2/5) During the 2001-03 seasons, the Cougar football teams were distinguished by three ten-win seasons, three top ten poll rankings, and appearances in the Sun, Rose, and Holiday Bowls. The Cougars shared the Pac-10 title in 2002. Alumnus Paul Wulff, WSU's 31st head coach, was fired in late November 2011, after compiling a 9-40 (.184) record in four seasons. Mike Leach coached the team from 2012 to 2019, guiding them to six bowl games. Leach left to coach Mississippi State in 2020 and was replaced by Hawaii head coach Nick Rolovich. Rolovitch was fired in October 2021 for not complying with the state's COVID-19 vaccine mandate. Defensive coordinator Jake Dickert was named the interim head coach.

**Thought:** The 2016 Washington State Cougars were led by Mike Leach. I need to search Mike Leach and find which other team he previously helmed.
**Action:** Search[Mike Leach]
**Observation:** Michael Charles Leach (March 9, 1961 – December 12, 2022) was an American college football coach who primarily coached at the NCAA Division I FBS level. He was a two-time national coach of the year, three-time conference coach of the year and the mastermind behind the NCAA record-setting air raid offense. He was the head coach at Texas Tech University from 2000 to 2009, where he became the winningest coach in school history. After Texas Tech, he coached at Washington State University from 2012 to 2019, where he recorded the third-most wins of any coach in school history. He then coached at Mississippi State from 2020 until his death in 2022.Leach was known for directing offenses using lots of passing to several receivers, in a spread system known as the air raid, which Leach developed with Hal Mumme when Mumme was head coach and Leach was offensive coordinator at Iowa Wesleyan, Valdosta State, and Kentucky in the 1990s. Leach's offenses with Mumme, and later as a head coach himself, have broken numerous school and NCAA records. On eighteen occasions, Leach led his unranked team to victory over a team ranked in the AP poll, which is the most in the AP era.

**Thought:** Mike Leach previously helmed Texas Tech University and Kentucky. So the answer is Texas Tech University and Kentucky.
**Action:** Finish[Texas Tech University and Kentucky]
**Observation:** Answer is INCORRECT

**Reflection:**

An example of the AlfWorld retrospective instruction prompt is shown below.

```
You will be given the history of a past experience in which you were placed in an environment and
given a task to complete. You were unsuccessful in completing the task. Do not summarize your
environment, but rather think about the strategy and path you took to attempt to complete the
task. Devise a concise, new plan of action that accounts for your mistake with reference to
specific actions that you should have taken. For example, if you tried A and B but forgot C, then
devise a plan to achieve C with environment-specific actions. You will need this later when you
are solving the same task. Give your plan after "Plan". Here are two examples:
```

**[Few-Shot Demonstrations]**

```
Your task is to: heat some mug and put it in coffeemachine.
> think: To solve the task, I need to find and take a mug, then heat it with stoveburner, then
put it in coffeemachine.
Ok.
> look
You are in the middle of a room. Looking quickly around you, you see nothing.
> think: First I need to find a mug. A mug is more likely to appear in countertop (1-2),
diningtable (1-2), drawer (1-3), fridge (1), sinkbasin (1), stoveburner (1-4), toaster (1). I can
check one by one, starting with countertop 1.
Ok.
> look
You are in the middle of a room. Looking quickly around you, you see nothing.
> go to countertop 1
On the countertop 1, you see a butterknife 1, a cup 1, a fork 2, and a mug 1.
> think: Now I find a mug (1). Next, I need to take it.
Ok.
> take mug 1 from countertop 1
You pick up the mug 1 from the countertop 1.
> think: Now I take a mug (1). Next, I need to heat it with stoveburner (1-4).
Ok.
> go to stoveburner 1
On the stoveburner 1, you see a pan 2.
> examine stoveburner 1
On the stoveburner 1, you see a pan 2.
> examine stoveburner 1
On the stoveburner 1, you see a pan 2.
> examine stoveburner 1
On the stoveburner 1, you see a pan 2.
> examine stoveburner 1
On the stoveburner 1, you see a pan 2.

STATUS: FAIL

New plan: I was stuck in a loop in which I continually examined stoveburner 1 instead of heating
mug 1 with stoveburner 1. I should have taken mug 1 from countertop 1, then heated it with
stoveburner 1, then put it in coffeemachine 1. It did not help to execute two identical actions
in a row. I will try to execute a different action if I am stuck in a loop again.
```

An example of the WebShop retrospective instruction prompt is shown below.

```
You will be given the history of a past experience in which you were placed in an environment and given a task
to complete. You were unsuccessful in completing the task. Do not summarize your environment, but rather think
about the strategy and path you took to attempt to complete the task. Devise a concise, new plan of action that
accounts for your mistake with reference to specific actions that you should have taken. There are two examples
below.

[Few-Shot Demonstrations]

Instruction:
i am looking for dairy free and apple variety pack of chips, and price lower than 30.00 dollars
[Search]

Action: search[dairy free and apple variety pack of chips]
Observation:
[Back to Search]
Page 1 (Total results: 50)
[Next >]
[B07HRFJWP8]
Enjoy Life Foods Soft Baked Ovals, Breakfast Bars, Nut Free Bars, Soy Free, Dairy Free, Non GMO, Gluten Free,
Vegan, Variety Pack, 4 Boxes (20 Bars Total)
$100.0
[B01KMHY5PG]
Enjoy Life Soft Baked Chewy Bars, Variety Pack, Nut Free Bars, Soy Free, Dairy Free, Gluten Free, 6 Boxes (30
Total Bars)
$21.49
[B008D2X8C4]
Enjoy Life Lentil Chips Variety Pack, Dairy Free Chips, Soy Free, Nut Free, Non GMO, Vegan, Gluten Free, 24
Bags (0.8 oz)
$100.0

Action: think[The first two items are not what I want. The third item is the dairy free and apple variety pack
of chips I want.]
Observation: OK.

Action: click[B008D2X8C4]
Observation:
[Back to Search]
[< Prev]
flavor name [dill and sour cream][garlic & parmesan][light sea salt][margherita pizza][thai chili lime][variety
pack]
size [0.8 ounce (pack of 24)][4 ounce (pack of 12)]
Price: $100.0
Rating: N.A.
[Description]
[Features]
[Reviews]
[Buy Now]

Action: think[The item has options 'variety pack' and '0.8 ounce (pack of 24)' and seems good to buy.]
Observation: OK.

Action: click[variety pack]
Observation: You have clicked variety pack.

Action: click[0.8 ounce (pack of 24)]
Observation: You have clicked 0.8 ounce (pack of 24).

Action: click[Buy Now]

STATUS: FAIL

Next plan: In this attempt, I was unsuccessful. I accidentally bought a product that was $100, which is more
than my budget of $30. Either way, the initial search results were not good. Next time, I will do
search["variety pack of chips"] and then check if the results meet the dairy free and the $30 budget
constraints. I will continue to refine my searches so that I can find more products.
```

