# OpenReview forum: "Retroformer: Retrospective Large Language Agents with Policy Gradient Optimization"
_ICLR.cc/2024/Conference — ICLR 2024 spotlight_

### Official Review · Reviewer_Hrx6 · 2023-10-15

**Soundness:** 3 good
**Presentation:** 3 good
**Contribution:** 2 fair
**Rating:** 6
**Confidence:** 3

**Summary:**

The paper propose a RLHF way to tune the prompt for the LLM agent.

**Strengths:**

The authors design a RLHF framework to tuning the prompt for the agents and have better scores on multiple tasks.

**Weaknesses:**

I think the main weakness is that this kind of prompt tuning may not be necessary for a well trained agent. The agent should be well tuned to understand all kinds of different prompts. So the problem itself may not be very significant. Instead of fixing the agents and tuning the prompts, tuning the agents from RLHF feedback may have a more significant effect for the LLM. But this is just my opinion. The authors can discuss whether it's correct or not.

**Questions:**

See the weakness section

---

> ### Author Response · Authors · 2023-11-22
> **Thanks for review - necessity of exploration, prompt tuning vs LLM tuning**
>
> Dear Reviewer Hrx6,
>
> We are sincerely grateful to the reviewer for the informative feedback, which has helped improve our paper (please see the updated paper and appendix.) Please see our point-to-point response below.
>
> > Q1: I think the main weakness is that this kind of prompt tuning may not be necessary for a well trained agent. The agent should be well tuned to understand all kinds of different prompts. So the problem itself may not be very significant.
>
> A1: Thanks for pointing it out! In light of your comments, we’ve added GPT-4, which has impressive decision-making capabilities, as the agents for comparison. Please see the updated Table 4, Fig. 4 and 6. In general, we find that even with GPT-4 as the agent, our Retroformer still shows significant improvements after just 1 episode. This is because even with a well-trained agent like GPT-4, it still needs exploration to cross off some incorrect answers and choose correct trajectories in the next round. If there is anything unclear about the updated results, please kindly let us know!
>
> > Q2: Instead of fixing the agents and tuning the prompts, tuning the agents from RLHF feedback may have a more significant effect for the LLM. But this is just my opinion. The authors can discuss whether it's correct or not.
>
> A2: Thanks for the question! In light of the comment, we’ve included one RL baseline, i.e., Soft Actor-Critic [1] that directly finetune the LLM agent online with environment feedback. For a fair comparison with the language agent methods in this paper, which all showed improvements in 5 episodes, we ran the RL method for 5 episodes but did not observe any improvement across all 3 environments. These results show that tuning the agent prompts turn out to be much more efficient than tuning the agent LLMs directly, under our problem setting.
>
> |     Method |     #Params |     #Retries |     HotPotQA |   |     AlfWorld |   |     WebShop |
> |------------|-------------|--------------|--------------|---|--------------|---|-------------|
> | SAC        |     2.25M   | N=1          | 27           |   | 58.95        |   | 30          |
> |            |             | N=4          | 27           |   | 59.7         |   | 30          |
>
> [1] Haarnoja, Tuomas, et al. "Soft actor-critic: Off-policy maximum entropy deep reinforcement learning with a stochastic actor." International conference on machine learning. PMLR, 2018.

---

> > ### Author Response · Authors · 2023-11-23
> > **Do the response and revision address your concern?**
> >
> > Dear Reviewer Hrx6,
> >
> > Thanks for your time dedicated to carefully reviewing this paper. We have tried to address your concerns in the response and updated submission - any feedback from you would be appreciated. If you have further comments, please kindly let us know - we hope for the opportunity to respond to them.
> >
> > Best wishes, Authors of paper 1478

---

### Official Review · Reviewer_SkxB · 2023-10-26

**Soundness:** 3 good
**Presentation:** 4 excellent
**Contribution:** 3 good
**Rating:** 8
**Confidence:** 4

**Summary:**

Summary: the paper introduces Retroformer, an algorithm for training a LLM to optimize a retrospective model which provides feedback on another "agent" model's trajectory, incorporating it into a prompt which the agent LLM can condition on for its next trial at the task.

They do this by rolling out an agent LLM in the environment (where observations and actions are all text), prompting the retrospective model on the trajectory and the final reward (which is computed heuristically), then prompting the retrospective model to output text which reflects on what went wrong and what the agent can do better next time. The actor model then conditions on this text.

They create many rollouts using this procedure, score them, and finetune the retrospective agent using PPO.

On HotPotQA, Alfworld, and Webshop, Retroformer shows modest success improvements over Reflexion, the previous SOTA.

**Strengths:**

- The paper is easy to read and positions itself clearly with respect to related work.
- It addresses a challenge which has not been solved yet - the problem of how to do RL finetuning with language agents.
- The proposed algorithm can be used in conjunction with a hosted actor LLM (e.g. GPT4) which cannot be finetuned. This seems important for making this algorithm useful for users.
- The results show consistent improvements over Reflexion.

**Weaknesses:**

- The paper's examples of bad/good prompts do not obviously show that the finetuned reflection model produces better prompts. For instance, in Figure 5, the Reflexion plan is better formatted but still doesn't point out that it was incorrect to return two series rather than one. It would be useful to see an analysis of how often the plan correctly identifies an improved plan (e.g. have a human hand-label 100 prompts for Reflexion and 100 from the frozen LM and record their success rates.)
-
- See my questions/suggestions below

**Questions:**

- I'm confused how the "f1 score" reward works.
- I'd like to see the following additional curves added to Fig 4 and 6 (possibly in an appendix), which might make it clearer how the finetuning contributes:
  - Retroformer, rolling out the base policy (before RL finetuning).  This would make it easier to see how much of the improvement above Reflexion is due to finetuning vs other algorithmic details.
  - Retroformer, rolling out the base policy, but with GPT-3 as the reflection model. This would answer the question of whether it's typically a better strategy to finetune a small model or just prompt a more capable cloud model.
- I'd recommend writing out the exact algorithm you used for PPO finetuning, especially since when PPO is used with language models slight details from the original algo are changed. Also, it's typically an online algorithm, so it would be helpful to detail any changes you made to make it work well in the offline setting.
- Equation 6 seems to be optimizing for each (x, y) pair individually (i.e. not treating a sequence of (x1, y1), (x2, y2) (x3, y3)... as a trajectory. Is this correct? If so, I'd recommend making this clearer in the text.

---

> ### Comment · Reviewer_SkxB · 2023-11-20
> **Correcting a typo**
>
> In my "Weaknesses" section, I realized I accidentally said "Reflexion" where I meant "Retroformer." Sorry for any confusion!

---

> ### Author Response · Authors · 2023-11-22
> **Thanks for review - correctness of plans, f1 score, two additional curves (Part I)**
>
> Dear Reviewer SkxB,
>
> We appreciate that you read our paper very carefully and your informative feedback, which has helped improve our paper. Below please see our response to your concerns. In case you find anything unclear in the paper, please kindly let us know.
>
> > Q1: It would be useful to see an analysis of how often the plan correctly identifies an improved plan, e.g. have a human hand-label 100 prompts for Reflexion and 100 from the frozen LM and record their success rates.
>
> A1: Thanks for the comment! Given the limited time, we’ve manually labeled 5 reflection responses in the HotPotQA dataset, including the “Teen Titans” examples mentioned in the review. We found that with human labeling of the root cause of failure and next plans, the agent LM can always succeed in the next trial. For the reflection responses from the frozen LM, i.e, the Reflexion method, 1 out of 5 tasks succeeds in the next trial and with our Retrofomer fine tuned by policy gradient, under the same content for the remaining prompt, 3 out of 5 tasks succeed.
>
> For the 100 tasks, one can refer to Figure 4. For example, under GPT-4, at episode 1, Reflexion has 48 successful tasks while Retroformer (r=4) has 51 successful tasks. This means that 8 (48-40) plans of frozen language model identifies an improved plan and 11 (51-40) plans of Retroformer identifies an improved plan, which shows the effectiveness of our approach.
>
> We appreciate your suggestion for hand labeling, and we will label (the rest of) the reflection responses and add them for fine tuning the Retrospective model, as an ablation model for human-in-the-loop expert demonstrations, similar to the Dagger RL method .
>
> [1] Ross, Stéphane, Geoffrey Gordon, and Drew Bagnell. "A reduction of imitation learning and structured prediction to no-regret online learning." Proceedings of the fourteenth international conference on artificial intelligence and statistics. JMLR Workshop and Conference Proceedings, 2011.
>
> > Q2: I'm confused how the "f1 score" reward works.
>
> A2: Thanks for spotting this! We’ve updated the Appendix C.3 to include the description of reward functions used in 3 environments. To answer your question, I quote the updated descriptions below.
>
> > F1 reward is used in the HotPotQA environment for comparing the matching of a generated answer to a question against the ground truth answer.  After removing the stopwords in both answers, we calculate the number of common tokens in two answers. Then Precision is # of common tokens divided by # of generated answer tokens and the Recall is # common tokens divided by # ground truth answer tokens. We can then compute f1 from precision and recall.
> This reward was defined in the ReAct paper, which is used as the baseline model in this paper.
>
> [2] Yao, Shunyu, et al. "React: Synergizing reasoning and acting in language models." arXiv preprint arXiv:2210.03629 (2022).
>
> > Q3-1: I'd like to see the following additional curves added to Fig 4 and 6 (possibly in an appendix), which might make it clearer how the finetuning contributes: Retroformer, rolling out the base policy (before RL finetuning). This would make it easier to see how much of the improvement above Reflexion is due to fine tuning vs other algorithmic details.
>
> A3-1: Thanks for the comment! In light of the suggestions, we’ve added the curve Retroformer (GPT-3, lora r=0) to Fig 4 and 6 to denote the base policy Retroformer agent with the frozen longchat-3b as the reflection model (before RL finetuning). The improvement due to finetuning is now entirely represented by the area, for example, of the curve Retroformer (GPT-3, lora r=4) above Retroformer (GPT-3, lora r=0).
>
> > Q3-2: I'd like to see the following additional curves. Retroformer, rolling out the base policy, but with GPT-3 as the reflection model. This would answer the question of whether it's typically a better strategy to finetune a small model or just prompt a more capable cloud model.
>
> A3-2: Thanks for the suggestions! The Reflexion (GPT-3) curve in Figure 4 and 6 is actually the requested “Retroformer, rolling out the base policy, but with GPT-3 as the reflection model”.(By the way, we believe you meant GPT-4 for the reflection model as a more capable cloud model; please let us know if we are wrong.)
> Therefore, We’ve also tried using GPT-4 as the reflection model, see Reflexion (GPT-4) and found that, for example, Retroformer (GPT-3/4, r=4) still outperformed it by a lot in all environments. This demonstrates that for generating reflective responses in a given environment, fine tuning a small model is more effective than prompting a more capable general-purpose model.

---

> > ### Comment · Reviewer_SkxB · 2023-11-23
> > **Score raised, one question remaining**
> >
> > Hi,
> >
> > Thanks for addressing my comments thoroughly! I especially appreciate the training-free curve in Fig 4 and 6 – this makes me a lot more confident that the finetuning of the retrospective model is helping.  As a result, I’ve raised my score.
> >
> > > The Reflexion (GPT-3) curve in Figure 4 and 6 is actually the requested “Retroformer, rolling out the base policy, but with GPT-3 as the reflection model”
> >
> > You're right I meant GPT4. I understand that the algorithms are identical, but I had assumed that your Reflexion baseline used the prompt from the Reflexion paper, and the Retroformer results used different prompts (which makes it hard to tell if the improvement comes from just better prompt tuning). If the prompts are different, then I still think adding the curve I suggested (equivalently, running Reflexion with the Retroformer prompt) would be good. If the prompts are identical, then this is great and makes your comparisons more convincing, so I’d recommend highlighting this explicitly. (Apologies if it’s already mentioned in the paper and I missed it).

---

> > > ### Author Response · Authors · 2023-11-23
> > > **Thanks for recommendation of accepting the paper and further comments**
> > >
> > > Dear Reviewer SkxB,
> > >
> > > Thank you for your encouraging feedback and recommendation of accepting the paper! Yes, the prompt template used by Retroformer and Reflexion is the same (see Appendix E). In the revised version, we will update the presentation to make it clearer.
> > >
> > > Thank you, Authors of submission SkxB

---

> ### Author Response · Authors · 2023-11-22
> **Thanks for review - PPO algorithm table,  equation 6 presentation (Part II)**
>
> > Q4: I'd recommend writing out the exact algorithm you used for PPO finetuning, especially since when PPO is used with language models slight details from the original algo are changed. Also, it's typically an online algorithm, so it would be helpful to detail any changes you made to make it work well in the offline setting.
>
> A4: Thanks for pointing it out! We’ve included an algorithm table for the PPO training in Appendix C.1 - Algorithm 1. We’ve also added a description of the three steps.
>
> > The offline PPO algorithm we used for finetuning the Retrospective component in this paper is presented below in \cref{Algo: PPO}. It contains three steps: offline data collection, reward model learning, and policy gradient finetuning. We use the offline ratings data to train a reward model first, and plug in the reward model for PPO finetuning
>
> > Q5: Equation 6 seems to be optimizing for each (x, y) pair individually (i.e. not treating a sequence of (x1, y1), (x2, y2) (x3, y3)... as a trajectory. Is this correct? If so, I'd recommend making this clearer in the text.
>
> A5: Thanks for carefully reading our paper! The presentation of this equation is actually correct.  $x$ in the equation refers to the instruction prompt, which includes textual descriptions of the agent trajectory, episodic returns, environment feedback at the end of one episode and $y$ denotes the reflective response generated by the Retrospective component taking this prompt as input. Therefore, for fine tuning the Retrospective model (Retroformer’s task), there is only one step in this model’s trajectory. In light of your comment, we’ve updated the presentation to make it clearer!
>
> > (note there is only 1 step in the Retrospective model’s trajectory in its episode)

---

### Official Review · Reviewer_96C3 · 2023-10-31

**Soundness:** 3 good
**Presentation:** 3 good
**Contribution:** 2 fair
**Rating:** 3
**Confidence:** 4

**Summary:**

This paper introduces "Retroformer", a framework designed to enhance LLM-assited agents. The Retroformer system comprises two language models, the actor and the retrospective model. The actor model executes tasks while the retrospective model provides feedback to the actor model, allowing it to self-improve. Retroformer employs both short-term and long-term memory to shape the rewards. Short-term memory is represented by the actor model's action history while long-term memory is created by appending summaries of previous failures to the task prompt. Experimental results demonstrates the effectiveness of Retroformer in various tasks.

**Strengths:**

- The overall framework designed by Retroformer is interesting and alleviates some of the shortcomings in previous works.
- The paper is well-structured with clear explanations of terminology and content, aiding in readability.

**Weaknesses:**

- The improvements brought by Retroformer are limited. There are no significant improvements in HotPotQA and WebShop, only meaningful improvement is observed in AlfWorld.
- The experiments are not solid enough. It lacks comparisons with RL methods that have been recognized to perform well within the same framework of interaction-feedback-learning. Additionally, there is no comparison with the currently acknowledged GPT-4 model, which has impressive decision-making capabilities, making it insufficient to demonstrate the contribution of this work.
- Only the prompt used in the HotPotQA environment is displayed, and it is difficult to determine whether special prompt engineering is needed in different environments. Therefore, it is insufficient to verify the claim of 'automatically tunes the language agent prompts'.

**Questions:**

- The feedback obtained from the interaction between the agent and the environment is usually sparse, and the computed rating $r=G_{k,i+1}-G_{k,i}$ represents the difference in return between two consecutive episodes. This means that the data used for finetuning retrospect models are not significantly different within a certain period. How does Retroformer address the overfitting issue caused by fine-tuning on a large amount of repetitive data?
- Are there any differences in the prompts used by Retroformer and baselines methods, and are there experimental results to analyze this part through ablation analysis?
- What are the possible reasons for limited performance improvements in HotPotQA and WebShop?
- How much efficiency is gained by using a retrospective model to tune prompts, rather than directly tuning the LLMs?
- Are there any details about the hyperparameters of PPO?

---

> ### Author Response · Authors · 2023-11-22
> **Thanks for the review - justification for performance improvement and RL baseline (Part I)**
>
> Dear Reviewer 96C3,
>
> We greatly appreciate your thorough and constructive comments, many of which have helped improve our paper. Both the paper and appendix have been updated extensively to include as much detail as possible. Please see our point-to-point response to your concerns below.
>
> > Q1: The improvements brought by Retroformer are limited. There are no significant improvements in HotPotQA and WebShop, only meaningful improvement is observed in AlfWorld.
>
> A1: Thanks for the comments. We’d like to discuss them in the following three aspects: (1) Improvements in HotPotQA, (2) Improvements in WebShop, and (3) Improvements against RL methods.
>
> **(1) Improvements in HotPotQA**
>
> We believe that there are significant improvements in HotpotQA by using Retroformer. For example, as in Table 2, Retrofomer (GPT-3, lora r=4) was improved by 14% in success rate within just one episode. The improvement is also 6% higher than the Reflexion (GPT-3) method that uses a frozen model without our proposed RLHF finetuning. In light of your comments, we’ve also tested GPT-4 as an agent in the updated experiments and found even GPT-4 gets stuck at 54% success rate. We believe that 54% is the upper bound of the success rate under existing LLMs.
> It deserves notice that HotPotQA, which is a text-based game, has an extremely large state and action space (you can treat it as a RL problem with a continuous action of 768 dimensions). It should take more than 500 episodes to observe certain improvements in some much simpler text-based environments in [1]. Our improvements (14%) by Retroformer in just one episode are significant.
>
> [1] Yuan, Xingdi, et al. "Counting to explore and generalize in text-based games." arXiv preprint arXiv:1806.11525 (2018).
>
> **(2) Improvements in WebShop**
>
> Thanks for carefully reading our paper! However, the difficulty of solving this web shop environment is well known in all recent language agent publications [2,3,4] that use this environment. As we stated in our previous submission, we agree that the performance improvement by Retroformer over the frozen baselines is observed but the improvements may be limited. This limitation of improvement was due to the fact that “web browsing requires a significant amount of exploration with more precise search queries”. Nevertheless, our Retroformer method is still performing better than Reflexion and React across episodes, which use a frozen model without our proposed RLHF fine tuning. In light of your comments, we’ve added GPT-4 agents in the updated experiments and found similar results in the updated Figure 6 (b).
>
> We’ve explained these results in the updated draft in Section 4 - Web Browsing.
> The results probably indicate that the verbal feedback approach (Reflexion, Retroformer) is not an optimal method for this environment, but our fine-tuning method still proves effective.
>
> [2] Yao, Shunyu, et al. "React: Synergizing reasoning and acting in language models." arXiv preprint arXiv:2210.03629 (2022).
> [3] Shinn, Noah, Beck Labash, and Ashwin Gopinath. "Reflexion: an autonomous agent with dynamic memory and self-reflection." arXiv preprint arXiv:2303.11366 (2023).
> [4] Liu, Xiao, et al. "Agentbench: Evaluating llms as agents." arXiv preprint arXiv:2308.03688 (2023).

---

> ### Author Response · Authors · 2023-11-22
> **Thanks for the review - performance improvement against RL baseline and under GPT-4 (Part II)**
>
> **(3) Improvements against RL methods**
>
> In light of your comments, to show the improvements against traditional RL methods, we added one method, i.e., Soft Actor-Critic [5] as the baseline.
> We have updated the experiment results in Table 2 and added the implementation details of the RL method in the updated Appendix C.2, which is quoted below.
> “We include one online RL algorithm, i.e., Soft Actor-Critic [5],  or SAC as a baseline model for comparison. Given that the three environments are text-based games, inspired by [1], we do mean-pooling for the embeddings of the generated text outputs, such as ``Search[It Takes a Family]'' as the agent actions. Therefore, the action space is continuous and is of 768 dimensions. We apply LoRA adapters with r=4 on the agent Action model instantiated from longchat-16k, and use SAC to do the online updates, with discount factor gamma=0.99, interpolation factor polyak=0.995, learning rate=0.01, entropy regularization alpha=0.2, and batch size=8.”
>
> For a fair comparison with the language agent methods in this paper, which all showed improvements in 5 episodes, we run SAC for 5 episodes but do not observe significant improvement across all 3 environments in the updated Table 2.
> |     Method |     #Params |     #Retries |     HotPotQA |   |     AlfWorld |   |     WebShop |
> |------------|-------------|--------------|--------------|---|--------------|---|-------------|
> | SAC        |     2.25M   | N=1          | 27           |   | 58.95        |   | 30          |
> |            |             | N=4          | 27           |   | 59.7         |   | 30          |
>
> This is mostly due to the fact that the three text-based games have an extremely large state and action space, and thus online exploration and learning within 5 episodes online cannot improve the LLM model with an extensive parameter count. Under this setting, verbal reinforcement with an LLM that is finetuned offline for generating effective reflective feedback (i.e., our Retroformer) is much more effective.
>
> To summarize for Q1, we do believe the improvements brought by Retroformer are not limited, especially if the reviewer considers how traditional RL methods perform under the same problem setting. We do observe significant improvements under HotPotQA and AlfWorld. Webshop requires a significant amount of exploration with more precise search queries but our Retroformer method still performs better than Reflexion and React across all episodes. The results under Webshop indicate that the verbal feedback approach (Reflexion, Retroformer) is not an optimal method for this environment, but our fine-tuning method still proves effective.  If there is anything unclear, please kindly let us know!
>
> [5] Haarnoja, Tuomas, et al. "Soft actor-critic: Off-policy maximum entropy deep reinforcement learning with a stochastic actor." International conference on machine learning. PMLR, 2018.
>
> > Q2: The experiments are not solid enough. It lacks comparisons with RL methods that have been recognized to perform well within the same framework of interaction-feedback-learning.
>
> A2: Thanks for raising the concern! Please refer to the detailed response above for the section Improvements against RL methods. In summary, we’ve added Soft Actor-Critic as the baseline. We run SAC for 5 episodes but do not observe significant improvement across all 3 environments with the updated results in Table 2.
> |     Method |     #Params |     #Retries |     HotPotQA |   |     AlfWorld |   |     WebShop |
> |------------|-------------|--------------|--------------|---|--------------|---|-------------|
> | SAC        |     2.25M   | N=1          | 27           |   | 58.95        |   | 30          |
> |            |             | N=4          | 27           |   | 59.7         |   | 30          |
>
> > Q3: Additionally, there is no comparison with the currently acknowledged GPT-4 model, which has impressive decision-making capabilities, making it insufficient to demonstrate the contribution of this work.
>
> A3: Thanks for pointing it out! In light of your comment, we have updated the paper draft by adding experiments using GPT-4  (See the updated Table 2, Fig. 4 and 6). In general, we observe consistent improvements at episode 0 where verbal feedback is not applied. This is because of the better decision-making capabilities of GPT-4. Furthermore, we still observe consistent improvements by Retroformer under GPT-4,  which again demonstrates the effectiveness of verbal feedback with a well-trained actor (e.g., GPT-4) together with a retrospective component tuned with our policy gradient method. We have updated the draft to include this analysis of GPT-4 results. If there is anything unclear, please kindly let us know!

---

> > ### Author Response · Authors · 2023-11-22
> > **Thanks for the review - prompt template, ratings, prompt difference, limitation reasons,  (Part III)**
> >
> > > Q4: Only the prompt used in the HotPotQA environment is displayed, and it is difficult to determine whether special prompt engineering is needed in different environments. Therefore, it is insufficient to verify the claim of 'automatically tunes the language agent prompts'.
> >
> > A4: Thanks for carefully reading our paper! In light of your comments, we have updated Appendix Section E to include all the prompts used in HotPotQA, Alfworld and Webshop environments. If there is anything unclear, please kindly let us know!
> >
> > > Q5: The feedback obtained from the interaction between the agent and the environment is usually sparse, and the computed rating, represents the difference in return between two consecutive episodes. This means that the data used for finetuning retrospect models are not significantly different within a certain period. How does Retroformer address the overfitting issue caused by fine-tuning on a large amount of repetitive data?
> >
> > A5: Thanks for the comment. We agree with the reviewer that the computed rating is usually sparse in online settings. This is exactly why we collect the data offline for fine-tuning the Retrospective component. It can also be found in our previous submission in Appendix, that the ratings in the offline dataset are not sparse. In fact, the instruction-response pairs with positive rating account for around 40% and the pairs with negative (sparse) ratings account for 60% in the training set. One could refer to our updated algorithm table in Appendix C.1 for the details on how we collected the data.
> >
> > > For HotPotQA environment, we collected 3,383 reflection samples by running the base rollout policy for 3 trials (N = 3) for 3,000 tasks in the training set, in which 1,084 instruction-response pairs have positive ratings. For AlfWorld, we collected 523 reflection samples and for WebShop, we collected 267 reflection samples.”
> >
> > Finally, The queries or task descriptions in the offline training set do not overlap with the ones we use to report the success rate (validation tasks) so there is no repetitive data or overfitting.
> >
> > > Q6: Are there any differences in the prompts used by Retroformer and baselines methods, and are there experimental results to analyze this part through ablation analysis?
> >
> > A6: Thanks for your question. No, there is no difference in the prompts used by Retroformer, ReAct and Reflexion. This is exactly why the three methods (ReAct, Reflextion, Retroformer) have the same success rate at episode 0. Reflextion is built upon ReAct by adding verbal feedback at the end of the episode. Retroformer further finetunes the Retrospective component by policy gradient. Therefore, the three language agent methods share the same prompt template.
> >
> > > Q7: What are the possible reasons for limited performance improvements in HotPotQA and WebShop?
> >
> > A7: Thanks for the questions. As we pointed out earlier, for complex text-based games, due to their extremely huge state-action space, one should not expect the method to solve the game within 5 episodes. For HotPotQA, we believe that 14% in success rate with one episode and 20% improvements within 5 episodes are significant performance improvements. For WebShop, it requires a significant amount of exploration with more precise search queries but our Retroformer method still performs better than Reflexion and React across episodes. The results under Webshop indicate that the verbal feedback approach (Reflexion, Retroformer) is not an optimal method for this environment, but our fine-tuning method still proves effective.  If there is anything unclear, please kindly let us know!

---

> ### Author Response · Authors · 2023-11-22
> **Thanks for the review - tuning prompt vs LLMs, hyperparameters (Part IV)**
>
> > Q8: How much efficiency is gained by using a retrospective model to tune prompts, rather than directly tuning the LLMs?
>
> A8: Thanks for the question. In light of your comments, we added one RL baseline (Soft Actor-Critic) in the updated experiments which directly tunes the LLMs instead of tuning the prompts. For a fair comparison with the language agent methods in this paper, which all showed improvements in 5 episodes, we run SAC for 5 episodes but do not observe improvement across all 3 environments. These results show that tuning the agent prompts are much more efficient than tuning the agent LLMs directly, under our problem setting.
> |     Method |     #Params |     #Retries |     HotPotQA |   |     AlfWorld |   |     WebShop |
> |------------|-------------|--------------|--------------|---|--------------|---|-------------|
> | SAC        |     2.25M   | N=1          | 27           |   | 58.95        |   | 30          |
> |            |             | N=4          | 27           |   | 59.7         |   | 30          |
>
> > Q9: Are there any details about the hyperparameters of PPO?
>
> A9: Thanks for pointing it out. We have updated in Appendix C.1 the details about the hyperparameters of PPO. We’ve also written out the exact algorithm we used for PPO finetuning in the updated Appendix C.1. Please kindly let us know if anything is not clear.
>
> > The offline PPO algorithm we used for finetuning the Retrospective component in this paper is presented below in Algorithm 1. It contains three steps: offline data collection, reward model learning, and policy gradient finetuning. We use the offline ratings data to train a reward model first, and plug in the reward model for PPO finetuning.

---

> > ### Comment · Reviewer_96C3 · 2023-11-23
> >
> > Thanks for the authors' response, however, I still have concerns about some issues.
> >
> > > about the sparse feedback
> >
> > I don't think collecting offline data fundamentally addresses the issue I raised. Although the case used by the authors is not that serious (but still sparse), in some more general and difficult tasks it's often more challenging to address the sparsity problem. I think Retroformer lacks a filter processing for the data.
> >
> > > about comparison with RL methods
> >
> > Considering that SAC is not a current SOTA method in RL, and the relatively modest performance difference observed on the WebShop task, I believe that the performance improvement offered by Retroformer might not be significant, especially when considering its substantial resource consumption compared to traditional methods. If possible, I would suggest the authors include more baselines to further validate this.
> >
> > > about comparison with GPT4
> >
> > The authors might have misunderstood my initial concern. I hope to see a full performance comparison between Retroformer and pure GPT4. My concern is that considering GPT4 has already shown enough excellent decision-making performance in some work and does not depend on extra feedback information, is it necessary to design a framework like Retroformer? Therefore, I want to see such a comparison to confirm this.

---

> > > ### Author Response · Authors · 2023-11-23
> > > **Thanks for further comments**
> > >
> > > Dear Reviewer 96C3,
> > >
> > > Thanks for your time spent on our submission and your further comments. We are glad that you now agree that the performance improvement on HotPotQA is significant.
> > >
> > > > About the sparse feedback
> > >
> > > Thanks for the comment! There is indeed a filtering processing for the data! One could refer to the Algorithm 1 - Step 1 in Appendix C.1. We always use a pair of reflective responses and label the response with higher ratings as the accepted response while the lower response is labeled as the rejected response. The stats reported in Data Collection is the number of positive/negative responses before filtering.
> > >
> > > This is the standard step for RLHF fine-tuning so we did explicitly not discuss it in text. One needs to filter and always will have a pair of (equal numbers of) accepted and rejected instruction response pairs in the end.
> > >
> > > One could refer to the RLHF reward model documentation for more details: https://huggingface.co/docs/trl/reward_trainer.
> > >
> > > In light of your comments, we will make this filtering process clearer in the revised draft.
> > >
> > > > About comparison with RL methods
> > >
> > > For a continuous RL control problem of an action of 768 dimensions, we believe it is impossible for any traditional RL methods to show substantial improvement in just one episode (~6 steps). Under this problem setting, verbal reinforcement is much more effective (and does not cost resource consumption, because it involves only model inference to generate the reflection; while RL methods need to tune significant numbers of model parameters online).
> > >
> > > If the reviewer could recommend a SOTA verbal reinforcement method, except for Reflexion that has been considered in this paper, please kindly let us know.
> > >
> > > > about comparison with GPT4
> > >
> > > Thanks for your comments. However, the updated ReAct (GPT-4) curve is indeed the pure GPT-4 performance. ReAct [1] is the algorithm used by AutoGPT and is the agent prompting method that is commonly adopted now both in academia and industry.
> > >
> > > [1] Yao, Shunyu, et al. "React: Synergizing reasoning and acting in language models." arXiv preprint arXiv:2210.03629 (2022).
> > >
> > > Best Regards, Authors of Submission 1478

---

> > > > ### Author Response · Authors · 2023-11-23
> > > > **Possible to provide your further feedback soon so we can reply?**
> > > >
> > > > Dear Reviewer 96C3,
> > > >
> > > > Thanks for your time and comments! Hope we are not bothering you, but we are looking forward to seeing whether our response properly addresses your comments and whether you have any further concerns, to which we hope for the opportunity to respond.
> > > >
> > > > We hope you will consider this work as an important step towards improving reflective autonomous language agent, especially in the real-world scenarios where only very few retries or exploration are allowed.
> > > >
> > > > With best regards,
> > > >
> > > > Authors of submission 1478

---

> > > > > ### Comment · Reviewer_96C3 · 2023-11-23
> > > > >
> > > > > Thanks for the author's response.
> > > > >
> > > > > - Regarding sparse feedback, my concern is that considering the small differences between consecutive episodes, even with processing such as retaining only higher rating data, they will still be very similar or even repeated. Thus, using episode return calculation as a rating design seems confusing to me.
> > > > > - Regarding comparison with RL methods, as in some recent LLM-based works [1], they compared various RL baselines. In [1], some RL methods [2,3] even outperformed other LLM baselines. Therefore, I think simply comparing with ReAct, Reflexion and SAC is not sufficient to demonstrate the superiority of Retroformer. However, considering the differences in experimental environments and tasks, the baselines I listed may not necessarily need to be compared. Instead, I suggest that the authors consider selecting a wider range of different types of baselines to fully demonstrate Retroformer's superiority.
> > > > >
> > > > > [1] SPRING: GPT-4 Out-performs RL Algorithms by Studying Papers and Reasoning.
> > > > >
> > > > > [2] Mastering diverse domains through world models.
> > > > >
> > > > > [3] Uncertainty-driven exploration for generalization in reinforcement learning.

---

> ### Author Response · Authors · 2023-11-23
> **Response to sparse feedback and further comparisons with additional RL methods**
>
> Dear Reviewer 96C3,
>
> Thanks for the prompt response. We are glad that we are aligned that the previous concerns on the comparisons with GPT4 have been resolved.
>
> 1. For the **sparse feedback problems**, let me try to explain this step by step to demonstrate why it is not sparse or repeated.
> For example, the agent was asked to answer a question in the HotpotQA by using Wikipedia APIs. In the first episode, it failed ($G_0=0$), and then multiple different reflections are generated, in which one reflection helps the agent succeed in the next round ($G_1=1$),; the other one doesn't help so the agent failed again in the next trial ($G_1=0$),. Then the first reflection is good and rated higher ($r=G_1-G_0=1$ and the second reflection is bad and rejected ($r=G_1-G_0=0$). The trajectory in the second episode is even not used for offline training in this example. Can the reviewer explain why the small differences between consecutive episodes or similarity could cause a problem for RLHF fine-tuning?
>
> 2. For continuous RL control problem, we are very confident that soft-actor critic is still the most stable SOTA baseline according to our knowledge. That is the reason why we chose it in the beginning during rebuttal. Furthermore, we do not believe any online traditional RL algorithm can learn within 1 episode (~6 steps) in these complex text-world environments (note **all state-of-the-art RL algorithms** including [2,3] in the paper [1] cited by reviewer have been trained for **1M steps**).  Could you please provide one specific RL baseline you would like us to try further, besides SAC, given the limited amount of time available?
>
> With best regards, Authors of submission 1478

---

> > ### Comment · Reviewer_96C3 · 2023-11-23
> >
> > - In the example you provide, there is a very clear reward, i.e., 1 for success and 0 for failure. However, in broader environments, providing such distinct feedback for an environment or task becomes challenging. In other words, more general environments often present situations like  $G_0=0.5, G_1=0.51$, resulting in $r=0.01$. And for most episodes data, the computed rating $r$ might consistently remain small, hindering RLHF's ability to effectively distinguish useful episodes. Consequently, this aspect of Retroformer's design poses a significant limitation, rendering it less suitable for general tasks.
> > - Considering the rebuttal deadline, it seems no more time to have further discussions. I can only suggest in a more general sense that the authors should consider comparing other kind of LLM-based methods or some exploration-based RL baselines (e.g., [1], [3] in the previous response, respectively) instead of limiting comparisons to refine-based methods like ReAct and Reflexion.

---

> > > ### Author Response · Authors · 2023-11-23
> > > **Thanks for your informative feedback!**
> > >
> > > Dear Reviewer 96C3,
> > >
> > > Thank you very much for the discussion, for which I hope we've resolved most concerns in original reviews and properly addressed some of the concerns raised during discussion.
> > >
> > > In this case, could you please consider updating your recommendation to reflect it (the original rating is 3)?
> > >
> > > As we are almost at time, we want to thank you again for your time spent on our discussion!
> > >
> > > With best regards, Authors of submission 1478

---

### Author Response · Authors · 2023-11-22
**[Last day for us to respond] Could you please go through and comment on our response?**

Dear Reviewers 96C3, SkxB and Hrx6,

Thanks for your time dedicated to carefully reviewing this paper.  It would be further highly appreciated if you let us know whether our response and the change in the paper properly address your concerns, despite your busy schedule. Thanks a lot!

Below let us summarize the new, informative experimental results inspired by your suggestions. As today is the last day for us to respond, we hope for the chance to see and respond to your feedback. Thank you very much!

**Updated presentation**

- To reviewer **96C3**: We have included all the prompts we used in three environments in the updated Appendix E.

- To reviewer **96C3**: The algorithm details of traditional RL agent (Soft Actor-Critic) are described in the updated Appendix C.2.

- To reviewer **96C3**: The offline data collection process is presented in the Algorithm table in in the updated Appendix C.1.

- To reviewers **96C3** and **SkxB**:  The exact algorithm of PPO and training hyperparameters are given in the updated Appendix C.1.

- To reviewer **SkxB**: Two requested curves Retroformer (GPT-4, LoRA r=4), Reflexion (GPT-4) were added to Fig. 4 and 6.

- To reviewer **SkxB**: We have made the reward functions (e.g., f1 score) clearer for each environment in Appendix C.3.

**Newly conducted experiments**

- To reviewers **96C3** and **Hrx6**: We have provided comparative results of our approach with traditional RL baseline (Soft Actor-Critic) that directly fine-tune the LLM agent on three environments  in Table 2, Figure 4 and 6. The results indicate that directly tuning LLMs using RL is inefficient compared with tuning prompts.

- To reviewers **96C3**, **SkxB** and **Hrx6**: We have conducted more thorough ablation experiments with Retroformer under GPT-4, a more capable cloud model for decision making. The results indicate that our fine-tuned Retrospective component still proves effective with a well-trained agent actor model, i.e., GPT-4.

- To reviewer **SkxB**: We have conducted the two ablation studies in light of your comments, to see how much of the improvement above Reflexion is due to fine tuning,  and to answer question whether it's typically a better strategy to finetune a small model or just prompt a more capable cloud model like GPT-4. Two additional curves are added.

With best regards, Authors of submission 1478

---

### Meta-Review · Area_Chair_UFcz · 2024-01-05

**Metareview:**

The authors study the problem of using LLM-based methods that integrate reasoning and tool usage (e.g., ReAct) for complex tasks requiring actions, proposing an 'actor' LLM that actually takes actions and generates response and a 'self-reflection' LLM that provides feedback to improve the actor LLM.  The specific technical contribution of this submission is to improve upon the verbal feedback mechanism utilized be recent self-reflection formulations (e.g., Reflexion) by directly refining the actor LLM prompts via policy gradient methods, thus being able to use fine-grained reward information while maintaining the generality of the 'self-reflection' framework (i.e., it can maintain assumptions regarding a frozen actor LLM). This 'Retroformer' model employs a combination of long-term memory (summaries of previous failures) and short-term memory (actor LLM action history) to shape the reward used by the prompt generating LLM. Empirical evaluation is performed on HotPotQA, AlfWorld, and WebShop -- showing solid performance improvements over strong recent, baselines.

Consensus strengths identified by reviewers regarding this submission include:
- The authors have identified a clear shortcoming of previous LLM 'self-reflection' works, namely relying exclusively on verbal feedback. To use reward from the environment to optimize the prompt generation policy is an intuitively appealing way to accomplish this.
- The paper is well-written and in particular contextualizes existing work within this vein very well. I was able to easily identify the precise contributions and how it contrasts with existing work.
- The resulting method is targeted at the preferred setup of keeping a frozen actor LLM, thus being general to many settings. Additionally, i believe that this specific configuration will facilitate more ML-algorithmic methodological contributions (i.e., it is a flexible formulation).
- The empirical results show consistent improvements over a strong, recent, well-received baseline approach.

Beyond the reviewer comments, I would also like to point out that:
- This is a very active area of research with clear academic interest and potential commercial impact.
- I think the novelty here is a non-trivial extension over Reflexion, etc. While the use of policy gradient isn't a leap, I think the specific formulation likely required a few iterations to actually get it to work. Additionally, it extends the decade+ long narrative of utilizing gradients based on environment/behavior feedback to derive a better end-to-end optimization.

Conversely, consensus limitations included:
- The exact examples shown in terms of improved prompts weren't totally convincing (and it is assumed these are the more useful examples). Really, this can only be addressed with a human-in-the-loop evaluation. This was partially addressed during rebuttal, but is still an area for future work (for which the authors have proposed a good plan).
- There were some questions regarding adding empirical configurations that better exemplify baselines (e.g., a particular configuration that is equivalent to using GPT-4 directly). However, this was well-addressed in rebuttal and is easily clarified in future revisions.
- Performance on WebShop is mixed, but in line with related approaches (i.e., there is still room for a 'breakthrough' here).
- One reviewer had concerns regarding the value of iterative improvements and effects of reward sparsity. However, I believe this was well-addressed in rebuttal.

Overall, this is a nice contribution in a very active research area of interest in both academic and industry settings. The prescribed approach is an 'expected' next step, but shown to work well and is almost certainly the state-of-the-art for ReAct-style formulations to solving multi-step problems with LLMs. I do think that the more negative reviewer has some valid concerns regarding directly tuning the actor LLM, but that is more of a philosophical debate beyond the scope of this paper. Additionally, the human-in-the-loop ablation study recommended would further elucidate the dynamics of Retroformer, especially how it contrasts with existing methods.

**Justification For Why Not Higher Score:**

While using policy gradient is sensible and shown to work well, it is also a natural extension -- even if I think it required some careful though to derive the specific formulation. Additionally, this conceptually and empirically builds on closely related work. Thus, it is a useful and well-executed, but also somewhat natural step in an existing line of work.

**Justification For Why Not Lower Score:**

This paper will almost certainly be of interest. The Reflexion paper that this builds on has been well-received and this is a clear conceptual and empirical improvement.

---

### Decision · Program_Chairs · 2024-01-16

Accept (spotlight)